# SANA: EFFICIENT HIGH-RESOLUTION IMAGE SYNTHESIS WITH LINEAR DIFFUSION TRANSFORMERS

**Enze Xie**[1*], **Junsong Chen**[1*], **Junyu Chen**[2,3], **Han Cai**[1], **Haotian Tang**[2],
**Yujun Lin**[2], **Zhekai Zhang**[2], **Muyang Li**[2], **Ligeng Zhu**[1], **Yao Lu**[1], **Song Han**[1,2]

[1]NVIDIA    [2]MIT    [3]Tsinghua University
Project Page: **nvlabs.github.io/Sana**

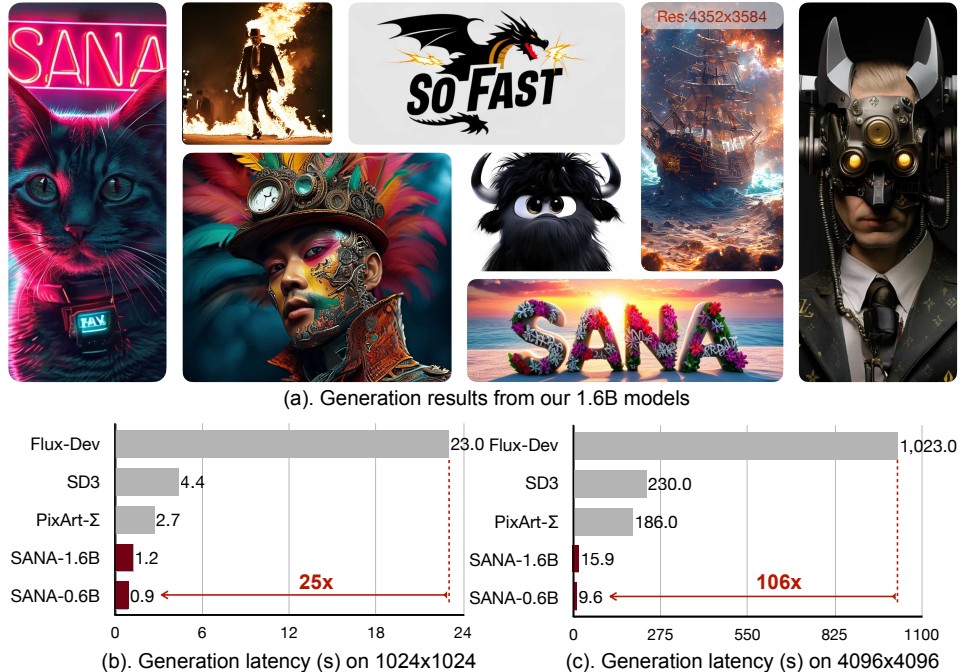

(a). Generation results from our 1.6B models

(b). Generation latency (s) on 1024x1024

(c). Generation latency (s) on 4096x4096

Figure 1: An overview of generated images and inference latency of Sana.

## ABSTRACT

We introduce Sana, a text-to-image framework that can efficiently generate images up to 4096×4096 resolution. Sana can synthesize high-resolution, high-quality images with strong text-image alignment at a remarkably fast speed, deployable on laptop GPU. Core designs include: (1) Deep compression autoencoder: unlike traditional AEs, which compress images only 8×, we trained an AE that can compress images 32×, effectively reducing the number of latent tokens. (2) Linear DiT: we replace all vanilla attention in DiT with linear attention, which is more efficient at high resolutions without sacrificing quality. (3) Decoder-only text encoder: we replaced T5 with modern decoder-only small LLM as the text encoder and designed complex human instruction with in-context learning to enhance the image-text alignment. (4) Efficient training and sampling: we propose Flow-DPM-Solver to reduce sampling steps, with efficient caption labeling and selection to accelerate convergence. As a result, Sana-0.6B is very competitive with modern giant diffusion model (e.g. Flux-12B), being 20 times smaller and 100+ times faster in measured throughput. Moreover, Sana-0.6B can be deployed on a 16GB laptop GPU, taking less than 1 second to generate a 1024×1024 resolution image. Sana enables content creation at low cost. Code and model will be publicly released.

---

*∗ Project co-lead.*

# 1 INTRODUCTION

In the past year, latent diffusion models have made significant progress in text-to-image research and have generated substantial commercial value. On one hand, there is a growing consensus among researchers regarding several key points: (1) Replace U-Net with Transformer architectures (Chen et al., 2024b;a; Esser et al., 2024; Labs, 2024), (2) Using Vision Language Models (VLM) for auto-labelling images (Chen et al., 2024b; OpenAI, 2023; Zhuo et al., 2024; Liu et al., 2024a) (3) Improving Variational Autoencoders (VAEs) and Text encoder (Podell et al., 2023; Esser et al., 2024; Dai et al., 2023) (4) Achieving ultra High-resolution image generation (Chen et al., 2024a), etc. On the other hand, industry models are becoming increasingly large, with parameter counts escalating from PixArt's 0.6B parameters to SD3 at 8B, LiDiT at 10B, Flux at 12B, and Playground v3 at 24B. This trend results in extremely high training and inference costs, creating challenges for most consumers who find these models difficult and expensive to use. Given these challenges, a pivotal question arises: *Can we develop a high-quality and high-resolution image generator that is computationally efficient and runs very fast on both cloud and edge devices?*

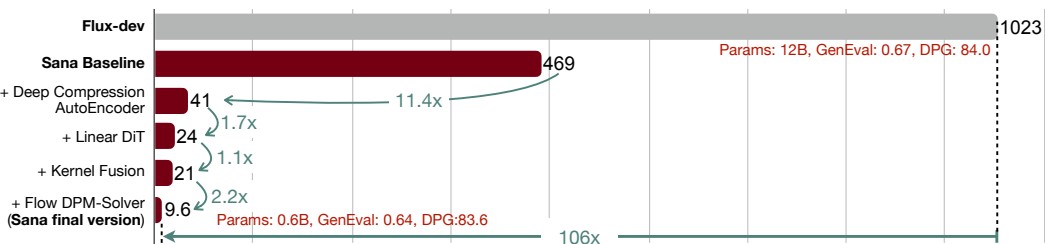

Figure 2: Algorithm and system co-optimize reduce the inference latency of 4096×4096 image generation, from 469 seconds to 9.6 seconds, and achieve **106×** faster than the current SOTA model, FLUX. The numbers are measured with batch size 1 on an A100 GPU.

This paper proposes Sana, a pipeline designed to efficiently and cost-effectively train and synthesize images at resolutions ranging from 1024×1024 to 4096×4096 with high quality. To our knowledge, no published works have directly explored 4K resolution image generation, except for PixArt-Σ (Chen et al., 2024a). However, PixArt-Σ is limited to generating images close to 4K resolution (3840×2160) and is relatively slow when producing such high-resolution images. To achieve this ambitious goal, we propose several core designs:

**Deep Compression Autoencoder:** We introduce a new Autoencoder (AE) in Section 2.1 that aggressively increases the scaling factor to 32. In the past, mainstream AEs only compressed the image's length and width with a factor of 8 (AE-F8). Compared with AE-F8, our AE-F32 outputs 16 × fewer latent tokens, which is crucial for efficient training and generating ultra-high-resolution images, such as 4K resolution.

**Efficient Linear DiT:** We introduce a new linear DiT to replace vanilla quadratic attention modules (Section 2.2). The computational complexity of the original DiT's self-attention is $O(N^2)$, which increases quadratically when processing high-resolution images. We replace all vanilla attention with linear attention, reducing the computational complexity from $O(N^2)$ to $O(N)$. At the same time, we propose Mix-FFN, which integrates 3×3 depth-wise convolution into MLP to aggregate the local information of tokens. We argue that linear attention can achieve results comparable to vanilla attention with proper design and is more efficient for high-resolution image generation (e.g., accelerating by 1.7× at 4K). Additionally, the indirect benefit of Mix-FFN is that we do not need position encoding (NoPE). For the first time, we removed the positional embedding in DiT and find no quality loss.

**Decoder-only Small LLM as Text Encoder:** In Section 2.3, we utilize the latest Large Language Model (LLM), Gemma, as our text encoder to enhance the understanding and reasoning capabilities regarding user prompts. Although text-to-image generation models have advanced significantly over the years, most existing models still rely on CLIP or T5 for text encoding, which often lack robust text comprehension and instruction-following abilities. Decoder-only LLMs, such as Gemma, exhibit strong text understanding and reasoning capabilities, demonstrating an ability to follow human instructions effectively. In this work, we first address the training instability issues that arise from directly adopting an LLM as a text encoder. Secondly, we design complex human instructions (CHI) to

leverage the LLM's powerful instruction-following, in-context learning, and reasoning capabilities to improve image-text alignment.

**Efficient Training and Inference Strategy:** In Section 3.1, we propose a set of automatic labelling and training strategies to improve the consistency between text and images. First, for each image, we utilize multiple VLMs to generate re-captions. Although the capabilities of these VLMs vary, their complementary strengths improve the diversity of the captions. In addition, we propose a clipscore-based training strategy (Section 3.2), where we dynamically select captions with high clip scores for the multiple captions corresponding to an image based on probability. Experiments show that this approach improve training convergence and text-image alignment. Furthermore, We propose a Flow-DPM-Solver that reduces the inference sampling steps from 28-50 to 14-20 steps compared to the widely used Flow-Euler-Solver, while achieving better results.

In conclusion, our Sana-0.6B achieves a throughput that is over $100\times$ faster than the current state-of-the-art method (FLUX) for 4K image generation (Figure 2), and $40\times$ faster for 1K resolution (Figure 4), while delivering competitive results across many benchmarks. In addition, we quantize Sana-0.6B and deploy it on an edge device, as detailed in Section 4. It takes only 0.37s to generate a $1024\times1024$ resolution image on a customer-grade 4090 GPU, providing a powerful foundation model for real-time image generation. We hope that our model can be efficiently utilized by all industry professionals and everyday users, offering them significant business value.

## 2 METHODS

### 2.1 DEEP COMPRESSION AUTOENCODER

#### 2.1.1 PRELIMINARY

To mitigate the excessive training and inference costs associated with directly running diffusion models in pixel space, Rombach et al. (2022) proposed latent diffusion models that operate in a compressed latent space produced by pre-trained autoencoders. The most commonly used autoencoders in previous latent diffusion works (Peebles & Xie, 2023; Bao et al., 2022; Cai et al., 2024; Esser et al., 2024; Dai et al., 2023; Chen et al., 2024b;a) feature a down-sampling factor of $F = 8$, mapping images from pixel space $\mathbb{R}^{H\times W\times 3}$ to latent space $\mathbb{R}^{\frac{H}{8}\times\frac{W}{8}\times C}$, where $C$ represents the number of latent channels. In DiT-based methods (Peebles & Xie, 2023), the number of tokens processed by the diffusion models is also influenced by another hyper-parameter, $P$, known as patch size. The latent features are grouped into patches of size $P \times P$, resulting in $\frac{H}{PF} \times \frac{W}{PF}$ tokens. A typical patch size in previous works is 2.

In summary, previous latent diffusion models (LDM), e.g. PixArt (Chen et al., 2024b), SD3 (Esser et al., 2024) and Flux (Labs, 2024), usually employ AE-F8C4P2 or AE-F8C16P2, where the AE compresses images by $8\times$ and DiT compresses by $2\times$. In our Sana, we aggressively scale the compression factor to $32\times$ and propose several techniques to maintain the quality.

#### 2.1.2 AUTOENCODER DESIGN PHILOSOPHY

Unlike the previous AE-F8, we aim to increase the compression ratio more aggressively. The motivation is that high-resolution images naturally contain more redundant information. Moreover, efficient training and inference of high-resolution images (e.g., 4K) necessitate a high compression ratio for the autoencoder. Table 1 illustrates that on MJHQ-30K, although previous methods (e.g., SDv1.5) have attempted to use AE-F32C64, the quality remains significantly inferior to that of AE-F8C4. Our AE-F32C32 effectively bridges this quality gap, achieving reconstruction capabilities comparable to SDXL's AE-F8C4. We believe that the minor difference in AE will not become a bottleneck for DiT's capability.

Moreover, instead of increasing the patch size $P$, we argue that the autoencoders should take full responsibility for compression, allowing the latent diffusion models to focus solely on denoising. Therefore, we develop an AE with a down-sampling factor of $F = 32$, Channel $C = 32$, and run diffusion models in its latent space with a patch size of 1 (AE-F32C32P1). This design reduces the number of tokens by $4\times$, significantly improving training and inference speed while lowering GPU memory requirements.

### 2.1.3 ABLATION OF AUTOENCODER DESIGNS

From the perspective of model structure, we implement several adjustments to accelerate convergence. Specifically, We replace the vanilla self attention mechanism with linear attention blocks to improve the efficiency of high-resolution generation. Additionally, from a training standpoint, we propose a multi-stage training strategy to improve training stability, which involving finetune our AE-F32C32 on $1024 \times 1024$ images to achieve better reconstruction results on high-resolution data.

**Can we compress tokens in DiT using a larger patch size?** We compare AE-F8C16P4, AE-F16C32P2 and AE-F32C32P1. These three settings compress a 1024×1024 image into the same number of token numbers $32 \times 32$. As shown in Figure 3(a), although AE-F8C16 exhibits the best reconstruction ability (rFID: F8C16<F16C32<F32C32), we empirically find that the generation results of F32C32 are superior (FID: F32C32P1<F16C32P2<F8C16P4).

Table 1: Reconstruction capability of different Autoencoders.

| Autoencoder | rFID ↓ | PSNR ↑ | SSIM ↑ | LPIPS ↓ |
|---|---|---|---|---|
| F8C4 (SDXL)[1] | 0.31 | 31.41 | 0.88 | 0.04 |
| F32C64 (SD)[2] | 0.82 | 27.17 | 0.79 | 0.09 |
| F32C32 (Ours) | 0.34 | 29.29 | 0.84 | 0.05 |

This indicates that allowing the autoencoder to focus solely on high-ratio compression and the diffusion model to concentrate on denoising is the optimal choice.

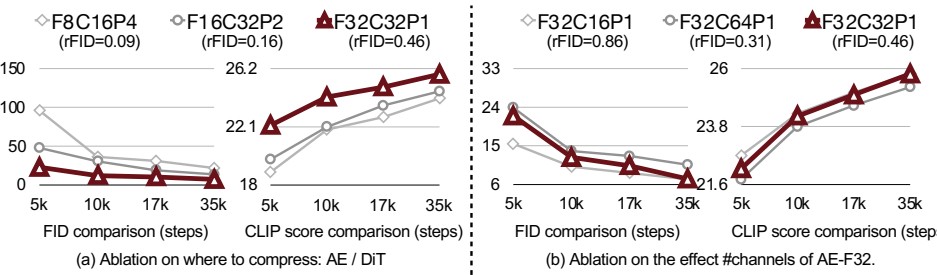

Figure 3: Ablation study on our deep compression autoencoder (AE).

**Different Channels in AE-F32**: We explore various channel configurations and finally choose C=32 as our optimal setting. As shown in Figure 3(b), fewer channels converge more quickly, but the reconstruction quality is worse. We observe that after 35K training steps, the convergence speeds of C=16 and C=32 are similar; however, C=32 yields better reconstruction metrics, resulting in better FID and CLIP scores. Although C=64 offers superior reconstruction, its following DiT's training convergence speed is significantly slower than that of C=32.

## 2.2 EFFICIENT LINEAR DiT DESIGN

The self-attention used by DiT has a computational complexity of $O(N^2)$, resulting in low computational efficiency when processing high-resolution images and incurring significant overhead. To address this issue, we first proposed linear DiT, which completely replaces the original self-attention with linear attention, achieving higher computational efficiency in high-resolution generation without compromising performance. In addition, we employ Mix-FFN to replace the original MLP-FFN, incorporating 3×3 depth-wise convolution to better aggregate token information. These micro designs are inspired by Cai et al. (2023); Xie et al. (2021), but we keep DiT's macro architecture design to maintain simplicity and scalability.

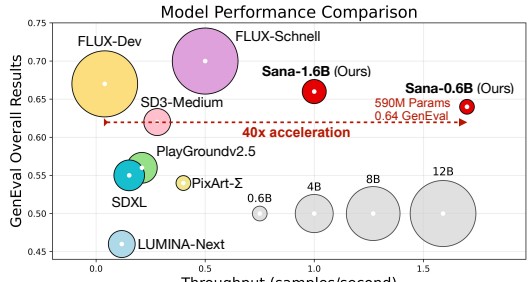

Figure 4: Comparison between Sana and state-of-the-art diffusion models under 1024×1024 resolution. All models are tested on an A100 GPU. Sana provides 0.64 GenEval overall performance with only 590M model parameters.

---

[1] https://huggingface.co/stabilityai/sdxl-vae
[2] https://github.com/CompVis/latent-diffusion

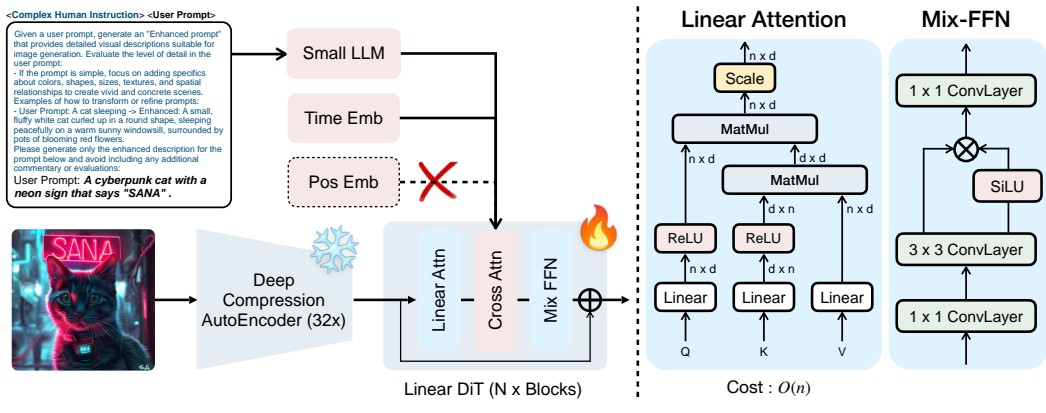

(a). Architecture overview of our Sana.    (b). Linear DiT Module.

Figure 5: **Overview of Sana**: Fig. (a) describes the high-level training pipeline, containing our $32\times$ deep compression Autoencoder, Linear DiT, and complex human instruction. Note that Positional embedding is not required in our framework. Fig. (b) describes the detailed design of the Linear Attention and Mix-FFN in Linear DiT.

**Linear Attention block.** An illustration of our utilized linear attention module is provided in Figure 5. To reduce computational complexity, we replace the traditional softmax attention with ReLU linear attention (Katharopoulos et al., 2020). While ReLU linear attention and other variants (Cai et al., 2023; Wang et al., 2020; Shen et al., 2021; Bolya et al., 2022) have primarily been explored in high-resolution dense prediction tasks, our work represents an early exploration to demonstrate the effectiveness of linear attention in image generation.

The computational benefits of our approach are evident in the implementation. As shown in Eq. 1, instead of computing attention for each query, we compute shared terms $\left(\sum_{j=1}^{N} \mathrm{ReLU}(K_j)^T V_j\right) \in \mathbb{R}^{d\times d}$ and $\left(\sum_{j=1}^{N} \mathrm{ReLU}(K_j)^T\right) \in \mathbb{R}^{d\times 1}$ only once. These shared terms can then be reused for each query, leading to a linear computational complexity of $O(N)$ in both memory and computation.

$$O_i = \sum_{j=1}^{N} \frac{\mathrm{ReLU}(Q_i)\mathrm{ReLU}(K_j)^T V_j}{\sum_{j=1}^{N} \mathrm{ReLU}(Q_i)\mathrm{ReLU}(K_j)^T} = \frac{\mathrm{ReLU}(Q_i)\left(\sum_{j=1}^{N} \mathrm{ReLU}(K_j)^T V_j\right)}{\mathrm{ReLU}(Q_i)\left(\sum_{j=1}^{N} \mathrm{ReLU}(K_j)^T\right)} \tag{1}$$

**Mix-FFN block.** As discussed in EfficientViT (Cai et al., 2023), linear attention models benefit from reduced computational complexity and lower latency compared to softmax attention. However, the absence of a non-linear similarity function may lead to sub-optimal performance. We observe a similar conclusion in image generation, where linear attention models suffer from much slower convergence despite eventually achieving comparable performance. To further improve training efficiency, we replace the original MLP-FFN with Mix-FFN. The Mix-FFN consists of an inverted residual block, a 3×3 depth-wise convolution, and a Gated Linear Unit (GLU) (Dauphin et al., 2017). The depth-wise convolution enhances the model's ability to capture local information, compensating for the weaker local information-capturing ability of ReLU linear attention. Performance ablations for the model design space are shown in Table 8.

**DiT without Positional Encoding (NoPE).** We are surprised that we can remove Positional Embedding without any loss in performance. Some earlier theoretical and practical works have mentioned that introducing 3×3 convolution with zero padding can implicitly incorporate the position information (Islam et al., 2020; Xie et al., 2021). In contrast to previous DiT-based methods that mostly use absolute PE, learnable PE, and RoPE, we propose NoPE, the first design that entirely omits positional embedding in DiT. Recent cutting-edge research in the LLM field (Kazemnejad et al., 2024; Haviv et al., 2022) has also indicated that NoPE may offer better length generalization ability.

**Triton Acceleration Training/Inference.** To further accelerate linear attention, we use Triton (Tillet et al., 2019) to fuse kernels for both the forward and backward passes of the linear attention blocks to speed up training and inference. By fusing all element-wise operations—including activation functions, precision conversions, padding operations, and divisions—into matrix multiplications, we reduce the overhead associated with data transfer. We attach more details and benefits coming from Triton to the appendix.

## 2.3 TEXT ENCODER DESIGN

**Why Replace T5 to Decoder-only LLM as Text Encoder?** The most advanced LLMs nowadays are decoder-only GPT architectures that are trained on a larger scale of data. Compared to T5 (a method proposed in 2019), decoder-only LLMs possess powerful reasoning capabilities. They can follow complex human instructions by using Chain-of-Thought (CoT) (Wei et al., 2022) and In-context-learning (ICL) (Brown, 2020). In addition, some small LLMs, such as Gemma-2 (Team et al., 2024a), can rival the performance of large LLMs while being very efficient. Therefore, we choose to adopt Gemma-2 as our text encoder.

As shown in Table. 9, Compared to T5-XXL, Gemma-2-2B has an inference speed that is 6× faster, while the performance of Gemma-2B is comparable to T5-XXL in terms of Clip Score, FID, GenEval, and HPSv2 (Wu et al., 2023).

**Stabilize Training using LLM as Text encoder:** We extract the last layer of features of the Gemma-2 decoder as text embedding. We empirically find that directly following the approach of using T5 text embedding as key, value, and image tokens (as the query) for cross-attention training results in extreme instability, with training loss frequently becoming NaN.

We find that the variance of T5's text embedding is several orders of magnitude smaller than that of the decoder-only LLMs (Gemma-1-2B, Gemma-2-2B, Qwen-2-0.5B), indicating that there are many large absolute values in the text embedding output. To address this issue, we add a normalization layer (i.e., RMSNorm) after the decoder-only text encoder, which normalizes the variance of the text embeddings to 1.0. In addition, we discover a useful trick that further accelerates model convergence by initializing a small learnable scale factor (e.g., 0.01) and multiplying it by the text embedding. The results are shown in Figure 6.

Table 2: Ablation study of whether using Complex Human Instruction (CHI).

| Prompt | Train Step | GenEval |
|---|---|---|
| User | 52K | 45.5 |
| CHI + User | | 47.7 (+2.2) |
| User | 140K | 52.8 |
| CHI + User | + 5K(finetune) | 54.8 (+2.0) |

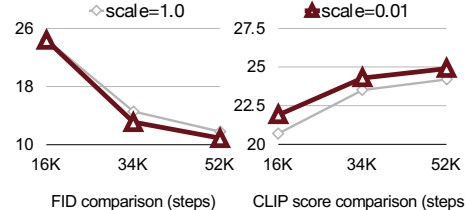

Figure 6: Ablation study of whether using text embedding normalization and small scale factor.

**Complex Human Instruction Improves Text-Image Alignment:** As mentioned above, Gemma has better instruction-following capabilities than T5. We can further leverage this capability to strengthen text embedding. Gemma is a chat model, and although it possesses strong capabilities, it can be somehow unpredictable, thus we need to add instructions to help it focus on extracting and enhancing the prompt itself. LiDiT (Ma et al., 2024) is the first to combine simple human instruction with user prompts. Here, we further expand it by using in-context learning of LLM to design a complex human instruction (CHI). As shown in Table 2, incorporating CHI during train—whether from scratch or through fine-tuning—can further improve the image-text alignment capability.

Additionally, as shown in Figure 7, we find that when given a short prompt such as "a cat", CHI helps the model generate more stable content. This is evident in the fact that models without CHI often output content unrelated to the prompt.

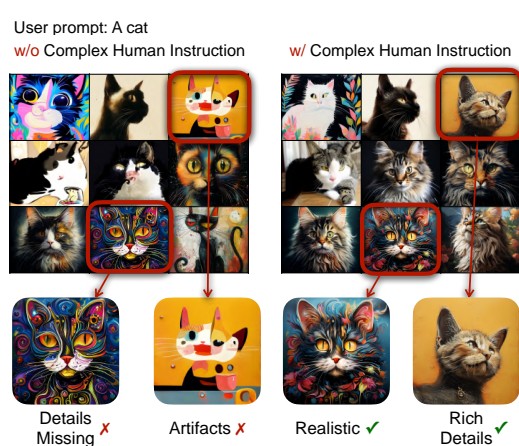

Figure 7: Generations w/ or w/o Complex-Human-Instruction (CHI). Without CHI, a simple prompt may lead to inferior generations, including artifacts and less-detailed results.

## 3 EFFICIENT TRAINING/INFERENCE

### 3.1 DATA CURATION AND BLENDING

**Multi-Caption Auto-labelling Pipeline:** For each image, whether or not it contains an original prompt, we will use four VLMs to label it: VILA-3B/13B (Lin et al., 2024), InternVL2-8B/26B (Chen et al., 2024d). Multiple VLMs can make the caption more accurate and more diverse.

**CLIP-Score-based Caption Sampler:** One problem multi-captioning presents is selecting the corresponding one for an image during training. The naive approach randomly selects a caption, which may select low-quality text and affect model performance.

We propose a clip score-based sampler, the motivation is to sample high-quality text with greater probability. We first calculate the clip score $c_i$ for all captions corresponding to an image, and then, when sampling, we sample according to the probability based on the clip score. Here, we introduce an additional hyper-parameter, temperature $\tau$, into the probability formulation $P(c_i) = \frac{\exp(c_i/\tau)}{\sum_{j=1}^{N} \exp(c_j/\tau)}$. The temperature can be used to adjust the sampling intensity. If the temperature is near 0, only the text with the highest clip score is sampled. The results in Table 4 show that variations in captions have minimal impact on image quality (FID) while improving semantic alignment during training.

**Cascade Resolution Training:** Benefiting from using AE-F32C32P1, we skip the 256px pre-training and start pre-training directly at a resolution of 512px, gradually fine-tuning the model to 1024px, 2K and 4K resolution. We believe that the traditional practice of pre-training at 256px is too cost-effective, as images at 256 resolution lose too much detailed information, resulting in slower learning for the model in terms of image-text alignment.

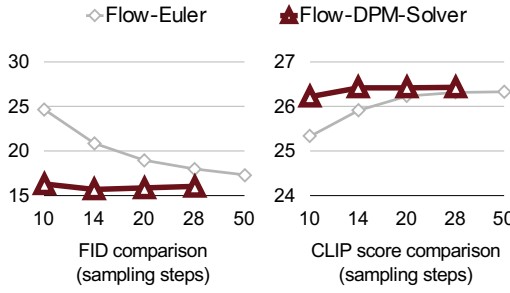

Figure 8: Impact of sampling steps on FID and CLIP-Score: A Comparison between Flow-DPM-Solver and Flow-Euler.

Table 3: Comparison of different training schedules on 256×256 resolution.

| Schedule | FID ↓ | CLIP ↑ | Iterations |
|---|---|---|---|
| DDPM | 19.5 | 24.6 | 120K |
| Flow Matching | 16.9 | 25.7 | 120K |

Table 4: Comparison of image-text pair sampling strategies during training.

| Prompt Strategy | FID ↓ | CLIP ↑ | Iterations |
|---|---|---|---|
| Single | 6.13 | 27.10 | 65K |
| Multi-random | 6.15 | 27.13 | 65K |
| Multi-clipscore | 6.12 | 27.26 | 65K |

### 3.2 FLOW-BASED TRAINING / INFERENCE

**Flow-based Training:** We analyze the superior performance of Rectified Flow from SD3 (Esser et al., 2024) and find that, unlike DDPM (Ho et al., 2020), which rely on noise prediction, both 1-Rectified Flow (RF) (Lipman et al., 2022) and EDM (Karras et al., 2022) use data or velocity prediction, resulting in faster convergence and improved performance. Specifically, all these methods follow a common diffusion formulation: $\mathbf{x}_t = \alpha_t \cdot \mathbf{x}_0 + \sigma_t \cdot \epsilon$, where $\mathbf{x}_0$ represents the image data, $\epsilon$ denotes random noise, and $\alpha_t$ and $\sigma_t$ are hyper-parameters of the diffusion process. In DDPM training, the objective is noise prediction, defined as $\epsilon_\theta(\mathbf{x}_t, t) = \epsilon_t$. However, both EDM and RF follow a different approach: EDM aims for data prediction with the objective $x_\theta(\mathbf{x}_t, t) = \mathbf{x}_0$, while RF uses velocity prediction with the objective $v_\theta(\mathbf{x}_t, t) = \epsilon - \mathbf{x}_0$. This transition from noise prediction to data or velocity prediction is critical near $t = T$, where noise prediction can lead to instability, while data or velocity prediction provides more precise and stable estimates. As noted by Balaji et al. (2022), attention activation near $t = T$ is stronger, further emphasizing the importance of accurate predictions at this critical point. This shift effectively reduces cumulative errors during sampling, resulting in faster convergence and improved performance. Further details can be found in Appendix B.

**Flow-based Inference**: In our work, we modify the original DPM-Solver++ (Lu et al., 2022b) adapting the Rectified Flow formulation, named Flow-DPM-Solver. The key adjustments involve substituting the scaling factor $\alpha_t$ with $1 - \sigma_t$, where $\sigma_t$ remains unchanged but time-steps are re-defined over the range $[0, 1]$ instead of $[1, 1000]$, with a time-step shift applied to achieve a lower signal-noise ratio, following SD3 (Esser et al., 2024). Additionally, our model predicts the velocity field, which differs from the data prediction in the original DPM-Solver++. Specifically, data is derived from the relation: data $\leftarrow x_0 = x_T - \sigma_T \cdot v_\theta(x_T, t_T)$, where $v_\theta(\cdot)$ is the velocity predicted by the model.

As a result, in Figure 8, our Flow-DPM-Solver converges at 14∼20 steps with better performance, while the Flow-Euler sampler needs 28∼50 steps for convergence with a worse result.

## 4 ON-DEVICE DEPLOYMENT

To enhance edge deployment, we quantize our model with 8-bit integers. Specifically, we adopt per-token symmetric INT8 quantization for activation and per-channel symmetric INT8 quantization for weights. Moreover, to preserve a high semantic similarity to the 16-bit variant while incurring minimal runtime overhead, we retain normalization layers, linear attention, and key-value projection layers within the cross-attention block at full precision.

Table 5: **On-device Deployment**: our inference engine with W8A8 quantization realized a 2.4× speedup when generating 1024px images on the laptop GPU. The performance of Sana is assessed with the CLIP-Score on MJHQ-30K (Li et al., 2024a) and the Image-Reward (Xu et al., 2024) on its first 1K images.

| Methods | Latency (s) | CLIP-Score ↑ | Image-Reward ↑ |
|---|---|---|---|
| Sana (FP16) | 0.88 | 28.5 | 1.03 |
| + W8A8 Quantization | 0.37 | 28.3 | 0.96 |

We implement our W8A8 GEMM kernel in CUDA C++ and employ kernel fusion techniques to mitigate the overhead associated with unnecessary activation loads and stores, thereby enhancing overall performance. Specifically, we integrate the $\text{ReLU}(K)^T V$ product of linear attention (Equation 1) with the $QKV$-projection layer; we also fuse the Gated Linear Unit (GLU) with the quantization kernel in Mix-FFN, and combine other element-wise operations. Additionally, we adjust the activation layout to avoid any transpose operations in GEMM and Conv kernels.

Table 5 shows the speed comparison before and after our deployment optimization on a laptop GPU. For generating a 1024px image, our optimized implementation achieves 2.4× speedup, taking only 0.37 seconds, while maintaining almost lossless image quality.

## 5 EXPERIMENTS

**Model Details.** Table 6 describes the details of the network architecture. Our Sana-0.6B only contains 590M parameters, and the number of layers and channels is almost identical to those of the original DiT-XL and PixArt-Σ. Our Sana-1.6B increases the parameters to 1.6B, with 20 layers and 2240 channels per layer, and increases the channels to 5600 in FFN. We believe that keeping the model layers between 20 and 30 strikes a good balance between efficiency and quality.

**Evaluation Details.** We use five mainstream evaluation metrics to evaluate the performance of our Sana, namely FID, Clip Score, GenEval (Ghosh et al., 2024), DPG-Bench (Hu et al., 2024), and ImageReward (Xu et al., 2024), comparing it with SOTA methods. FID and Clip Score are evaluated on the MJHQ-30K (Li et al., 2024a) dataset, which contains 30K images from Midjourney. GenEval and DPG-Bench both focus on measuring text-image alignment, with 533 and 1,065 test prompts, respectively. ImageReward assesses human preference performance and includes 100 prompts.

Table 6: Architecture details of the proposed Sana.

| Model | Width | Depth | FFN | #Heads | #Param (M) |
|---|---|---|---|---|---|
| Sana-0.6B | 1152 | 28 | 2880 | 36 | 590 |
| Sana-1.6B | 2240 | 20 | 5600 | 70 | 1604 |

Table 7: **Comprehensive comparison of our method with SOTA approaches in efficiency and performance.** The speed is tested on one A100 GPU with FP16 Precision. Throughput: Measured with batch=16. Latency: Measured with batch=1 and sampling step=20. We highlight the **best**, second best, and *third best* entries.

| Methods | Throughput (samples/s) | Latency (s) | Params (B) | Speedup | FID ↓ | CLIP ↑ | GenEval ↑ | DPG ↑ |
|---|---|---|---|---|---|---|---|---|
| **512 ×512 resolution** | | | | | | | | |
| PixArt-α (Chen et al., 2024b) | 1.5 | 1.2 | 0.6 | 1.0× | 6.14 | 27.55 | 0.48 | 71.6 |
| PixArt-Σ (Chen et al., 2024a) | 1.5 | 1.2 | 0.6 | 1.0× | *6.34* | *27.62* | *0.52* | *79.5* |
| **Sana-0.6B** | 6.7 | 0.8 | 0.6 | 5.0× | 5.67 | 27.92 | 0.64 | 84.3 |
| **Sana-1.6B** | 3.8 | 0.6 | 1.6 | 2.5× | **5.16** | **28.19** | **0.66** | **85.5** |
| **1024 ×1024 resolution** | | | | | | | | |
| LUMINA-Next (Zhuo et al., 2024) | 0.12 | 9.1 | 2.0 | 2.8× | 7.58 | 26.84 | 0.46 | 74.6 |
| SDXL (Podell et al., 2023) | 0.15 | 6.5 | 2.6 | 3.5× | 6.63 | 29.03 | 0.55 | 74.7 |
| PlayGroundv2.5 (Li et al., 2024a) | 0.21 | 5.3 | 2.6 | 4.9× | *6.09* | **29.13** | 0.56 | 75.5 |
| Hunyuan-DiT (Li et al., 2024c) | 0.05 | 18.2 | 1.5 | 1.2× | 6.54 | 28.19 | 0.63 | 78.9 |
| PixArt-Σ (Chen et al., 2024a) | 0.4 | 2.7 | 0.6 | 9.3× | 6.15 | 28.26 | 0.54 | 80.5 |
| DALLE3 (OpenAI, 2023) | - | - | - | - | - | - | 0.67 | 83.5 |
| SD3-medium (Esser et al., 2024) | 0.28 | 4.4 | 2.0 | 6.5× | 11.92 | 27.83 | 0.62 | 84.1 |
| FLUX-dev (Labs, 2024) | 0.04 | 23.0 | 12.0 | 1.0× | 10.15 | 27.47 | 0.67 | *84.0* |
| FLUX-schnell (Labs, 2024) | 0.5 | 2.1 | 12.0 | 11.6× | 7.94 | 28.14 | **0.71** | **84.8** |
| **Sana-0.6B** | 1.7 | 0.9 | 0.6 | 39.5× | 5.81 | 28.36 | 0.64 | 83.6 |
| **Sana-1.6B** | 1.0 | 1.2 | 1.6 | 23.3× | **5.76** | *28.67* | *0.66* | **84.8** |

Table 8: **Performance of Sana block design space.** The speed is tested on one A100 GPU with FP16 Precision with 1024 image size. MACs: Multi-accumulate operations for a single forward pass. TP (Throughput): Measured with batch=16. Latency: Measured with batch=1.

| Blocks | AE | MACs (T) | TP (/s) | Latency (ms) |
|---|---|---|---|---|
| FullAttn & FFN | F8C4P2 | 6.48 | 0.49 | 2250 |
| + LinearAttn | F8C4P2 | 4.30 | 0.52 | 1931 |
| + MixFFN | F8C4P2 | 4.19 | 0.46 | 2425 |
| + Kernel Fusion | F8C4P2 | 4.19 | 0.53 | 2139 |
| LinearAttn & MixFFN | F32C32P1 | 1.08 | 1.75 | 826 |
| +Kernel Fusion | F32C32P1 | 1.08 | 2.06 | 748 |

## 5.1 PERFORMANCE COMPARISON AND ANALYSIS

We compare Sana with the most advanced text-to-image diffusion models in Table 7. For $512 \times 512$ resolution, Sana-0.6 demonstrates a throughput that is 5× faster than PixArt-Σ, which has a similar model size, and significantly outperforms it in FID, Clip Score, GenEval, and DPG-Bench. For $1024 \times 1024$ resolution, Sana is considerably stronger than most models with <3B parameters and excels in inference latency. Our models achieve competitive performance even when compared to the most advanced large model FLUX-dev. For instance, while the accuracy on DPG-Bench is equivalent and slightly lower on GenEval, Sana-0.6B's throughput is 39× faster, and Sana-1.6B is 23× faster.

In Table 8, we analyze the efficiency of replacing the original DiT's modules with the corresponding linear DiT's modules under the $1024 \times 1024$ resolution setting. We observe that using AE-F8C4P2, replacing the original full attention with linear attention can reduce latency from 2250ms to 1931ms, but the generation results are worse. Replacing the original FFN with our Mix-FFN compensates for the performance loss, although it sacrifices some efficiency. With Triton kernel fusion, our linear DiT can ultimately be slightly faster than the original DiT at the 1024px scale and faster at higher resolution. Moreover, when upgrading from AE-F8C4P2 to AE-F32C32P1, the MACs can be further reduced by 4×, and throughput can also be improved by 4×. Triton kernel fusion can bring ∼10% speed acceleration.

The left side of Figure 9 compares the generation results of Sana, Flux-dev, SD3, and PixArt-Σ. In the first row of text rendering, PixArt-Σ lacks text rendering capability, while Sana can render text accurately. In the second row, the quality of the images generated by our Sana and FLUX is comparable, while SD3's text understanding is inaccurate. The right side of Figure 9 shows that Sana can be successfully deployed on a laptop locally. A Demo video is provided in the appendix.

Table 9: **Comparison of different Text-Encoders.** All models are tested with an A100 GPU with FP16 precision after 50K training steps. Gemma-2B models achieve better performance than T5-large at a similar speed and comparable performance to the larger, much slower T5-XXL.

| Text-Encoder | #Params (M) | Latency (s) | FID ↓ | CLIP ↑ | GenEval ↑ | HPSv2 ↑ |
|---|---|---|---|---|---|---|
| T5-XXL | 4762 | 1.61 | 6.1 | 27.1 | 0.51 | 27.95 |
| T5-Large | 341 | 0.17 | 6.1 | 26.2 | 0.39 | 27.57 |
| Gemma2-2B | 2614 | 0.28 | 6.0 | 26.9 | 0.45 | 27.92 |
| Gemma-2B-IT | 2506 | 0.21 | 5.9 | 26.8 | 0.47 | 27.78 |
| Gemma2-2B-IT | 2614 | 0.28 | 6.1 | 26.9 | 0.50 | 27.92 |

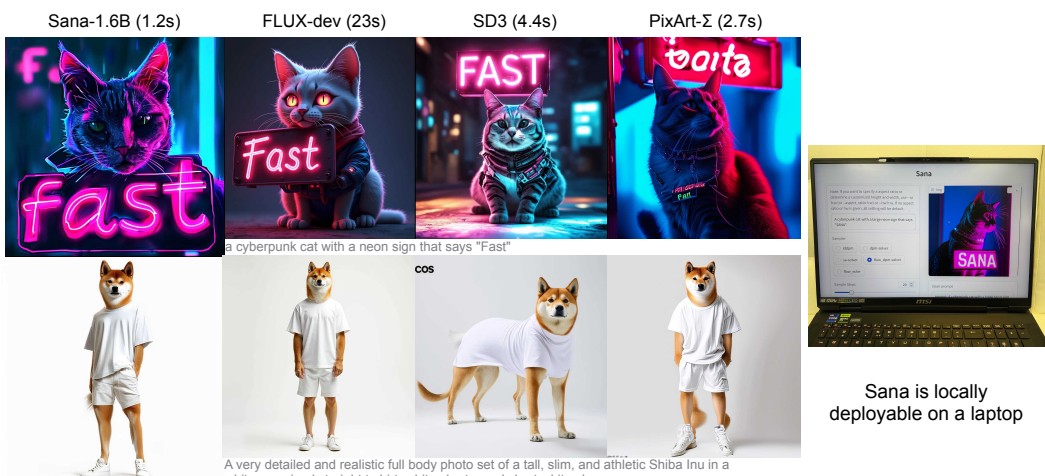

Figure 9: **Left**: Visualization comparison of Sana-1.6B vs FLUX-dev, SD3 and PixArt-Σ. The speed is tested on an A100 GPU with FP16 precision. **Right**: Quantize Sana-1.6B is deployable on a GPU laptop generating an 1K×1K image within 1 seconds.

# 6 RELATED WORK

We put a relatively brief overview of related work in the main text, with a more comprehensive version available in the supplementary material. In terms of generative model architecture, Diffusion Transformer (Peebles & Xie, 2022) and DiT-based Text-to-image extensions (Chen et al., 2024b; Esser et al., 2024; Labs, 2024) have made significant progress over the past year. Regarding text encoder, the earliest work (Rombach et al., 2022) uses CLIP, while subsequent works (Saharia et al., 2022; Chen et al., 2024b;a) adopt T5-XXL. There are also efforts that combine T5 and CLIP (Balaji et al., 2022; Esser et al., 2024). For high-resolution generation, PixArt-Σ (Chen et al., 2024a) is the first model capable of directly generating images at 4K resolution. Additionally, GigaGAN (Kang et al., 2023) can generate 4K images using a super-resolution model. In the context of on-device deployment, Zhao et al. (2023); Li et al. (2024b) have explored the deployment of diffusion models on mobile devices.

# 7 CONCLUSION

This paper builds a new efficient text-to-image pipeline named Sana. We have made improvements in the following dimensions: we propose a deep compression autoencoder, widely use linear attention to replace self-attention in DiT, utilize decoder-only LLM as text encoder with complex human instruction, establish an automatic image caption pipeline, and propose flow-based DPM-Solver to accelerate sampling. Sana can generate images at a maximum resolution of 4096×4096, delivering throughput more than 100× higher than the SOTA methods while maintaining competitive generation results.

In the future, we will consider building an efficient video generation pipeline based on Sana. A potential limitation of this work is that it cannot fully guarantee the safety and controllability of the generated image content. Additionally, challenges remain in certain complex cases, such as text rendering and the generation of faces and hands.

**Acknowledgements.** We would like to thank Shuchen Xue from UCAS, Cheng Lu from OpenAI, Jincheng Yu from HKUST, and Chongjian Ge from HKU for discussions and data collection.

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

## A    FULL RELATED WORK

**Efficient Diffusion Transformers.** The introduction of Diffusion Transformers (DiT) (Peebles & Xie, 2023) marked a significant shift in image generation models, replacing the traditional U-Net architecture with a transformer-based approach. This innovation paved the way for more efficient and scalable diffusion models. Building upon DiT, PixArt-$\alpha$ (Chen et al., 2024b) extended the concept to text-to-image generation, demonstrating the versatility of transformer-based diffusion models. Stable Diffusion 3 (SD3) (Esser et al., 2024) further advanced the field by proposing the Multi-modal Diffusion Transformer (MM-DiT), which effectively integrates text and image modalities. More recently, Flux (Labs, 2024) showcased the potential of DiT architectures in high-resolution image generation by scaling up to 12B parameters. In addition, earlier works like CAN (Cai et al., 2024) and DiG (Zhu et al., 2024) explored linear attention mechanisms in class-condition image generation. Several works are also related to modifying the model configuration, e.g., diffusion without attention (Yan et al., 2024; Teng et al., 2024) and cascade model structures (Pernias et al., 2023; Ren et al., 2024; Tian et al., 2024).

**Text Encoders in Image Generation.** The evolution of text encoders in image generation models has significantly impacted the field's progress. Initially, Latent Diffusion Models (LDM) (Rombach et al., 2022) adopted OpenAI's CLIP as the text encoder, leveraging its pre-trained visual-linguistic representations. A paradigm shift occurred with the introduction of Imagen (Saharia et al., 2022), which employed the T5-XXL language model as its text encoder, demonstrating superior text understanding and generation capabilities. Subsequently, eDiff-I (Balaji et al., 2022) proposed a hybrid approach, ensemble T5-XXL, and CLIP encoders to combine their respective strengths in language comprehension and visual-textual alignment. Recent advancements (Ma et al., 2024; Liu et al., 2024b; Hu et al., 2024; Liu et al., 2024a), such as Playground v3, have explored the use of decoder-only Large Language Models (LLMs) as text encoders, potentially offering more nuanced text understanding and generation. This trend towards more sophisticated text encoders reflects the ongoing pursuit of improved text-to-image alignment and generation quality in the field.

**On Device Deployment.** Several studies have explored post-training quantization (PTQ) techniques to optimize diffusion model inference for edge devices. Research in this area has focused on calibration objectives and data acquisition methods. BRECQ (Li et al., 2021) incorporates Fisher information into the objective function. ZeroQ (Cai et al., 2020) uses distillation to generate proxy input images for PTQ. SQuant (Guo et al., 2022) employs random samples with objectives based on Hessian spectrum sensitivity. Recent work such as Q-Diffusion (Li et al., 2023) has achieved high-quality generation using only 4-bit weights. In our work, we choose W8A8 to reduce peak memory usage.

## B    MORE IMPLEMENTATION DETAILS

**Rectified-Flow vs. DDPM.** In our theoretical analysis, we investigate the reasons behind the fast convergence of flow-matching methods, demonstrating that both 1st flow-matching and EDM models rely on similar formulations. Unlike DDPMs, which use noise prediction, flow-matching and EDM focus on data or velocity prediction, resulting in improved performance and faster convergence. This shift from noise prediction to data prediction is particularly critical at $t = T$, where noise prediction tends to be unstable and leads to cumulative errors. As noted by Balaji et al. (2022), attention activation near $t = T$ grows stronger, highlighting the necessity of accurate predictions at this key moment in the sampling process.

As discussed in Lu (2023), the behavior of diffusion models near $t = T$ reveals that when t $\approx$ T, the data distribution resembles noise, and noise prediction approaches randomness. The challenge arises because the errors made at $t = T$ propagate through all subsequent sampling steps, making it crucial for the sampler to be particularly precise near this time step. Based on Tweedie's formula, the gradient of the log density at time $t$, $\nabla_{\mathbf{x}_t} \log q_t(\mathbf{x}_t)$, is approximated by:

$$\nabla_{\mathbf{x}_t} \log q_t(\mathbf{x}_t) = -\frac{\mathbf{x}_t - \alpha_t \mathbb{E}_{q_{0t}(\mathbf{x}_0|\mathbf{x}_t)}[\mathbf{x}_0]}{\sigma_t^2}. \tag{2}$$

When $t \approx T$, $\mathbf{x}_0$ and $\mathbf{x}_t$ become conditionally independent, leading to $q_{0t}(\mathbf{x}_0 \mid \mathbf{x}_t) \approx q_0(\mathbf{x}_0)$. Consequently, the noise prediction model's optimal solution becomes:

$$\epsilon_\theta(\mathbf{x}_t, t) \approx -\sigma_t \nabla_{\mathbf{x}_t} \log q_t(\mathbf{x}_t) \approx \frac{\mathbf{x}_t - \alpha_t \mathbb{E}_{q_0(\mathbf{x}_0)}[\mathbf{x}_0]}{\sigma_t}. \tag{3}$$

Since $\mathbb{E}_{q_0(\mathbf{x}_0)}[\mathbf{x}_0]$ is independent of $\mathbf{x}_t$, the noise prediction model simplifies to a linear function of $\mathbf{x}_t$. However, as discussed in Section 5.2.1, this additional linearity can result in more accumulated errors during sampling, explaining why the original DPM-Solver struggles with guided sampling in such cases.

To address this issue and improve stability, DPM-Solver (Lu et al., 2022a) proposes modifying the noise prediction model to a more stable parameterization. By subtracting all linear terms inspired by equation 3, the remaining term is proportional to $\mathbb{E}_{q_0(\mathbf{x}_0)}[\mathbf{x}_0]$, corresponding to the data prediction model. Specifically, when $t \approx T$, the data prediction model approximates a constant:

$$\mathbf{x}_\theta(\mathbf{x}_t, t) \approx \frac{\mathbf{x}_t + \sigma_t^2 \nabla_{\mathbf{x}_t} \log q_t(\mathbf{x}_t)}{\alpha_t} \approx \mathbb{E}_{q_0(\mathbf{x}_0)}[\mathbf{x}_0]. \tag{4}$$

Thus, for $t \approx T$, the data prediction model becomes approximately constant, and the discretization error for integrating this constant is significantly smaller than for the linear noise prediction model. This insight guides our development of an improved Flow-DPM-Solver based on DPM-Solver++ (Lu et al., 2022b), which adapts a velocity prediction model Sana to a data prediction one, enhancing performance for guided sampling.

**Flow-based DPM-Solver Algorithm.** We present the rectified flow-based DPM-Solver sampling process in Algorithm 1. This modified algorithm incorporates several key changes: hyper-parameter and time-step transformations, as well as model output transformations. These adjustments are highlighted in blue to differentiate them from the original solver.

In addition to improvements in FID and CLIP-Score, which are shown in Figure 8 of the main paper, our Flow-DPM-Solver also demonstrates superior convergence speed and stability compared to the Flow-Euler sampler. As illustrated in Figure 10, Flow-DPM-Solver retains the strengths of the original DPM-Solver, converging in only 10-20 steps to produce stable, high-quality images. By comparison, the Flow-Euler sampler typically requires 30-50 steps to reach a stable result.

---

**Algorithm 1** Flow-DPM-Solver (Modified from DPM-Solver++)

---

**Require:** initial value $x_T$, time steps $\{t_i\}_{i=0}^M$, data prediction model $x_\theta$, velocity prediction model $v_\theta$, time-step shift factor $s$

1: Denote $h_i := \lambda_{t_i} - \lambda_{t_{i-1}}$ for $i = 1, \ldots, M$

2: $\tilde{\sigma}_{t_i} = \frac{s \cdot \sigma_{t_i}}{1 + (s-1) \cdot \sigma_{t_i}}$, $\alpha_{t_i} = 1 - \tilde{\sigma}_{t_i}$            ▷ Hyper-parameter and Time-step transformation

3: $x_\theta(\tilde{x}_{t_i}, t_i) = \tilde{x}_{t_i} - \tilde{\sigma}_{t_i} v_\theta(\tilde{x}_{t_i}, t_i)$                      ▷ Model output transformation

4: $\tilde{x}_{t_0} \leftarrow x_T$. Initialize an empty buffer $Q$.

5: $Q_{\text{buffer}} \leftarrow x_\theta(\tilde{x}_{t_0}, t_0)$

6: $\tilde{x}_{t_1} \leftarrow \frac{\tilde{\sigma}_{t_1}}{\tilde{\sigma}_{t_0}} \tilde{x}_{t_0} - \alpha_{t_1} \left(e^{-h_1} - 1\right) x_\theta(\tilde{x}_{t_0}, t_0)$

7: $Q_{\text{buffer}} \leftarrow x_\theta(\tilde{x}_{t_1}, t_1)$

8: **for** $i = 2$ to $M$ **do**

9:      $r_i \leftarrow \frac{h_{i-1}}{h_i}$

10:      $D_i \leftarrow \left(1 + \frac{1}{2r_i}\right) x_\theta(\tilde{x}_{t_{i-1}}, t_{i-1}) - \frac{1}{2r_i} x_\theta(\tilde{x}_{t_{i-2}}, t_{i-2})$

11:      $\tilde{x}_{t_i} \leftarrow \frac{\tilde{\sigma}_{t_i}}{\tilde{\sigma}_{t_{i-1}}} \tilde{x}_{t_{i-1}} - \alpha_{t_i} \left(e^{-h_i} - 1\right) D_i$

12:      **if** $i < M$ **then**

13:          $Q_{\text{buffer}} \leftarrow x_\theta(\tilde{x}_{t_i}, t_i)$

14:      **end if**

15: **end for**

16: **return** $\tilde{x}_{t_M}$

---

**Multi-Caption Auto-labeling Pipeline.** In Figure 12, we present the results of our CLIP-Score-based multi-caption auto-labeling pipeline, where each image is paired with its original prompt and 4 captions generated by different powerful VLMs. These captions complement each other, enhancing semantic alignment through their variations.

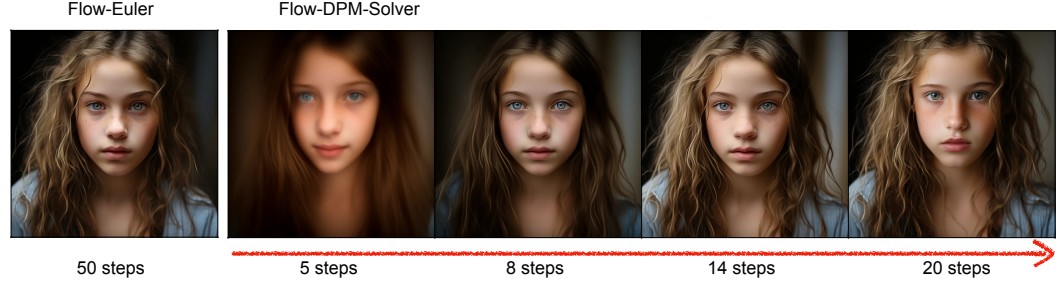

Figure 10: Visual comparison of Flow-Euler Sampler with 50 steps and Flow-DPM-Solver with 5/8/14/20 steps.

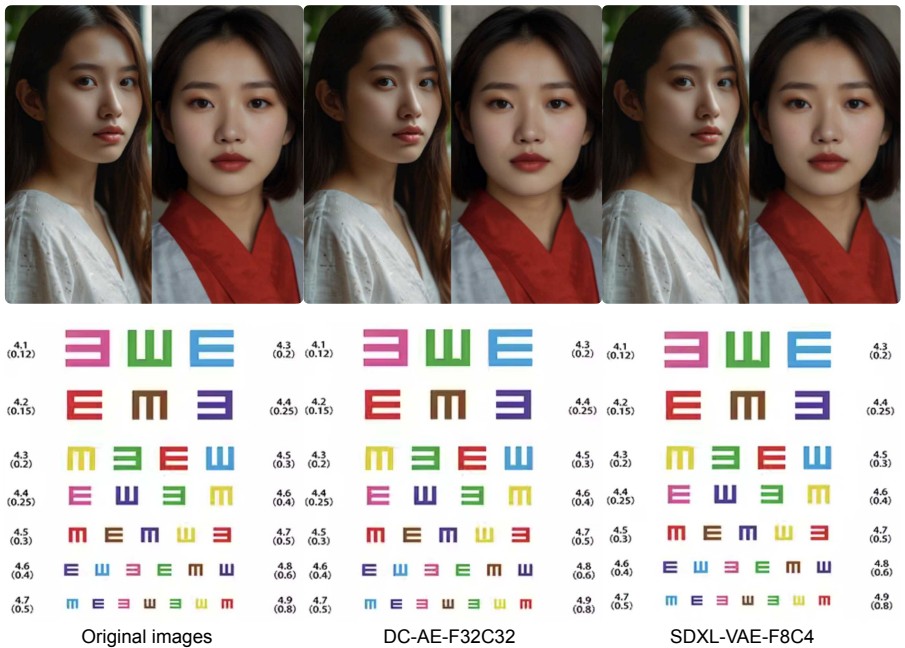

Figure 11: Comparison between the original images, our DC-AE-F32C32 (Chen et al., 2024c) and SDXL's VAE-F8C4.

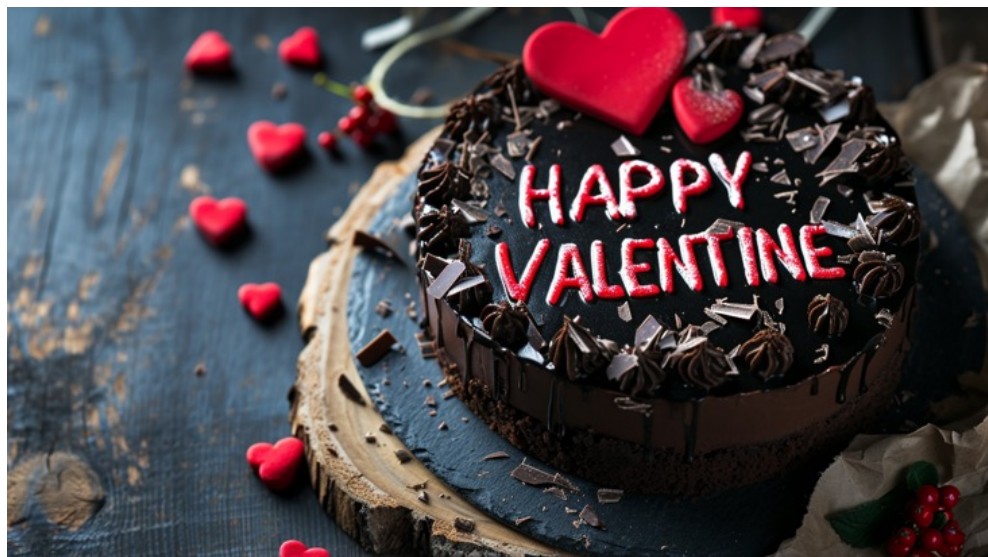

Caption_original (ClipScore: 25.67)

top view the written " HAPPY VALENTINE " on a tart chocolate cake, black wood background

Caption_VILA-13B (ClipScore: 26.33)

The image captures a delightful scene of a Valentine's Day celebration. At the center of the image is a round chocolate cake, rich and inviting. The cake is adorned with a red heart on top, symbolizing love and affection. The words "Happy Valentine's" are elegantly written in red icing on the cake, adding to the festive atmosphere. The cake is placed on a wooden table, which provides a rustic charm to the setting. Scattered around the cake are red heart-shaped confetti, further enhancing the Valentine's theme. In the background, there's a hint of a bouquet of red roses, a classic symbol of love and romance. The roses are slightly blurred, drawing focus to the cake in the foreground. Overall, the image beautifully encapsulates the spirit of Valentine's Day, with its focus on love, affection, and celebration.

Caption_VILA-3B (ClipScore: 25.46)

The image captures a delightful scene of a Valentine's Day celebration. At the center of the frame is a round chocolate cake, its surface adorned with a glossy chocolate ganache. The cake is further embellished with a scattering of chocolate shavings and a heart-shaped topper, adding to its festive appeal. The cake is placed on a wooden surface, which provides a rustic contrast to the rich, dark color of the chocolate. The background is dark, allowing the cake to stand out prominently. Adding to the Valentine's Day theme, red heart-shaped sprinkles are scattered around the cake, their vibrant color popping against the dark backdrop. The cake is positioned slightly off-center, drawing the viewer's attention to it. The overall composition of the image suggests a carefully arranged celebration, with the cake as the star of the show. The image does not contain any discernible text. The relative positions of the objects and their detailed descriptions provide a comprehensive understanding of the image content.

Caption_InternVL2-26B (ClipScore: 25.57)

The image features a beautifully decorated chocolate cake placed on a dark wooden surface. The cake is richly covered in a layer of dark chocolate ganache, giving it a smooth and glossy appearance. The top of the cake is adorned with piped chocolate swirls and heart- shaped decorations, adding a touch of elegance and romance. At the center of the cake, the words "HAPPY VALENTINE" are written in bold, red icing, creating a striking contrast against the dark chocolate background. Surrounding the cake, there are additional heart- shaped decorations in red, some of which are placed on the wooden surface, while others are scattered around the cake. The overall setting is warm and inviting, with a rustic wooden background that complements the rich, indulgent nature of the chocolate cake. The use of red hearts and the "HAPPY VALENTINE" message clearly indicate that this cake is intended for a Valentine's Day celebration

Caption_InternVL2-8B (ClipScore: 26.37)

The image features a beautifully decorated chocolate cake with a heartfelt Valentine's Day message. The cake is round and sits on a dark, rustic wooden surface. The cake is covered in a rich, dark chocolate ganache, and it is adorned with a generous amount of chocolate shavings and crushed nuts, giving it a luxurious and indulgent appearance. At the top of the cake, the words "HAPPY VALENTINE" are written in bold, red icing, creating a striking contrast against the dark chocolate. The icing is piped in a slightly uneven, handwritten style, adding a personal touch to the message. Surrounding the cake are several red heart-shaped decorations, some of which are whole and others that are broken into smaller pieces, scattered around the cake. These hearts add to the romantic and festive theme of the cake. The cake is placed on a dark, round cake board, which complements the rustic wooden background. The overall presentation is elegant and festive, making it perfect for a Valentine's Day celebration

Figure 12: Illustration of re-caption of an image with multiple VLMs.

**Discussion: CLIP Text Encoder's Input Token Length limitation.** Sana primarily uses the CLIP model to score the caption as the selective strategy during training. The 77-token limitation of the original CLIP model may seem restrictive. However, after extensive observation, we found that the most valuable content typically appears in the first half of the caption, such as "The image captures **$content of image$**...", that is, 77-token length is typically sufficient to encapsulate essential semantic information for our purposes. Consequently, we believe that the CLIP-Score based on 77

tokens still offers valuable insights and relevance for assessing caption quality in our context. In the future, we will attempt to fine-tune a larger context-length CLIP model or use more advanced VLM to score the text-image pairs.

**Triton Acceleration Training/Inference Detail.** This section describes how to accelerate the training inference with kernel fusion using Triton. Specifically, for the forward pass, the ReLU activations are fused to the end of QKV projection, the precision conversions and padding operations are fused to the start of KV and QKV multiplications, and the divisions are fused to the end of QKV multiplication. For the backward pass, the divisions are fused to the end of the output projection, and the precision conversions and ReLU activations are fused to the end of KV and QKV multiplications.

**2K/4K Fine-tuning.** We reintroduce Positional Encoding (PE) to improve performance during the fine-tuning of 2K models on top of 1K models. For fine-tuning 4K models based on 2K models, we apply the same PE interpolation strategy used in Chen et al. (2024a). The training process with the addition of positional encoding (PE) converges remarkably quickly, typically within just 10K iterations.

**Positional Embedding.** As previously discussed, positional embeddings can be effectively re-implemented in just a few training steps. Specifically, our experiments demonstrate that the model rapidly adapts to the absolute positional embedding, utilizing the sin-cos formulation, through a brief fine-tuning process. This adaptation occurs within merely 10K training steps, using a total batch size of 1024.

**Definition of Gemma2-2B-IT.** Gemma2 (Team et al., 2024b) is a family of lightweight, state-of-the-art open models from Google, built from the same research and technology used to create the Gemini models. The specific Gemma we use in Sana is the Gemma-2-2B-IT (Team, 2024) version. "IT" here represents the model is Instruct-Tuned for better prompt following and chat ability, experiments show that instruction-following tuned versions are superior to non-Instruction-Tuned models, as shown in Table 9.

## C  MORE RESULTS

**Comparison Between Autoencoders** Figure 11 illustrates the visual differences between the original images and the reconstructions generated by two distinct models: our DC-AE-F32C32 (Chen et al., 2024c) and SDXL's VAE-F8C4. Both models deliver reconstructions nearly indistinguishable from the original images, showcasing their powerful encoding and decoding capabilities.

**Ablation on Sana Blocks.** Table 13 describes how different block designs affect performance. Directly switching from DiT's self-attention to linear attention will result in FID and Clip Score performance loss, but adding Mix-FFN can compensate for the performance loss. Adding triton kernel fusion can speed up training/inference without negatively impacting performance.

**Compare SANA's CHI with LiDiT's SHI.** LiDiT (Ma et al., 2024) is the first work that attempts to promote the reasoning ability of LLM through relatively simple human instruction(SHI). Additionally, LiDiT is necessary to fine-tune the last few layers of the LLM model, which increases the training cost and pipeline complexity. In contrast, our CHI designs more complex prompt templates and utilizes in-context-learning techniques, which can fully stimulate LLM's high-order reasoning ability, extract better text embedding, and thus improve semantic alignment capabilities.

**Complex Human Instruction Analysis.** To observe the effectiveness of CHI, we input the user prompt with/without CHI into Gemma-2. We believe a strong positive correlation exists between LLM output and text embedding quality. As shown in Figure 13, without CHI, although Gemma-2 can understand the meaning of the input, the output is conversational and does not focus on understanding the user

Table 10: Comparison of CHI vs. SHI performance on GenEval metric during fine-tuning.

| Training Step | CHI | SHI |
|---|---|---|
| +2K steps | 0.636 | 0.617 |
| +5K steps | 0.642 | 0.626 |

prompt itself. After adding CHI, Gemma-2's output is better focused on understanding and enhancing the details of the user prompt. Besides, as illustrated in Table 10, we fine-tune both CHI and SHI

settings from a pre-trained model without human instructions. After 2K and 5K fine-tuning steps, CHI consistently shows better GenEval performance.

**Detailed Results on DPG-Bench, GenEval and ImageReward.** As an extension of Table. 7 in the main paper, we show all the metric details of GenEval, DPG-Bench, and ImageReward for reference in Table 11 and Table 12 respectively.

**Finding: Zero-shot Language Transfer Ability.** As shown in Figure 14, we were surprised to find that by using Gemma-2 as the text encoder and Chinese/Emoji expressions as text prompts; our Sana can also understand and generate corresponding images. Note that we filter out all prompts other than English during training, so Gemma-2 brings the zero-shot generation capability of Chinese/Emoji.

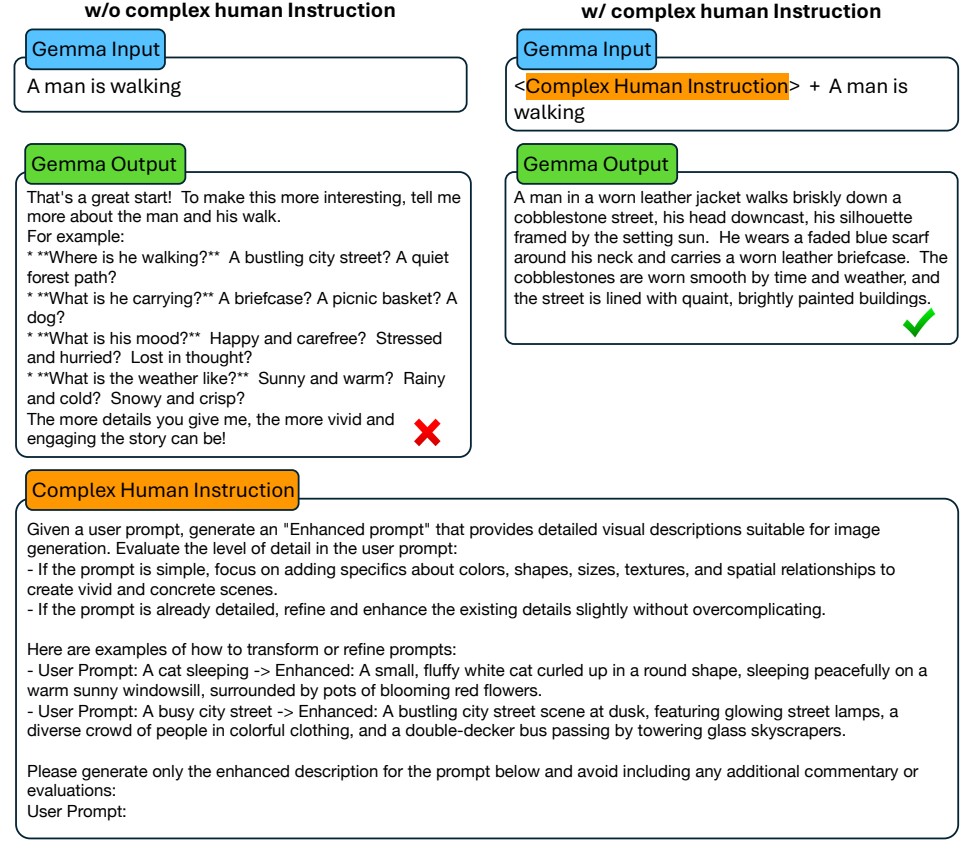

Figure 13: Illustration of Gemma-2's output with/without complex human instruction, and the full prompt of our complex human instruction.

**Detailed Speed Comparison of Text Encoder.** In Table 14, we compare the latency and parameters for various T5 models alongside the Gemma models. Notably, the Gemma-2B model exhibits a similar latency to T5-large while significantly increasing the model size. This enhancement in model size is a key factor in achieving improved capabilities with greater efficiency.

**Detailed Speed Comparison of Diffusion Model.** In Table 15, we compare the throughput and latency of the mainstream DiT-based text-to-image method and our model in detail and test them at resolutions of 512, 1024, 2048, and 4096, respectively. Our Sana is far ahead of other methods at different resolutions. As the resolution increases, the efficiency advantage of our Sana becomes more significant.

**More Visualization Images.** As shown in Figure 15, we can see that 4K images can directly generate more details than 1k images. In Figure 16, we show more images generated by our model with various prompts. We also provide a video mp4 demo in the supplementary material (zip file) to show that Sana is deployed on a laptop.

Table 11: **Comparison of SOTA methods on GenEval with details.** The table includes different metrics such as overall performance, single object, two objects, counting, colors, position, and color attribution.

| Model | Params (B) | Overall ↑ | Objects | | Counting | Colors | Position | Color Attribution |
| --- | --- | --- | --- | --- | --- | --- | --- | --- |
| | | | Single | Two | | | | |
| **512 ×512 resolution** | | | | | | | | |
| PixArt-α | 0.6 | **0.48** | 0.98 | 0.50 | 0.44 | 0.80 | 0.08 | 0.07 |
| PixArt-Σ | 0.6 | **0.52** | 0.98 | 0.59 | 0.50 | 0.80 | 0.10 | 0.15 |
| **Sana-0.6B** (Ours) | 0.6 | **0.64** | 0.99 | 0.71 | 0.63 | 0.91 | 0.16 | 0.42 |
| **Sana-1.6B** (Ours) | 0.6 | **0.66** | 0.99 | 0.79 | 0.63 | 0.88 | 0.18 | 0.47 |
| **1024 ×1024 resolution** | | | | | | | | |
| LUMINA-Next (Zhuo et al., 2024) | 2.0 | **0.46** | 0.92 | 0.46 | 0.48 | 0.70 | 0.09 | 0.13 |
| SDXL (Podell et al., 2023) | 2.6 | **0.55** | 0.98 | 0.74 | 0.39 | 0.85 | 0.15 | 0.23 |
| PlayGroundv2.5 (Li et al., 2024a) | 2.6 | **0.56** | 0.98 | 0.77 | 0.52 | 0.84 | 0.11 | 0.17 |
| Hunyuan-DiT (Li et al., 2024c) | 1.5 | **0.63** | 0.97 | 0.77 | 0.71 | 0.88 | 0.13 | 0.30 |
| DALLE3 (OpenAI, 2023) | - | **0.67** | 0.96 | 0.87 | 0.47 | 0.83 | 0.43 | 0.45 |
| SD3-medium (Esser et al., 2024) | 2.0 | **0.62** | 0.98 | 0.74 | 0.63 | 0.67 | 0.34 | 0.36 |
| FLUX-dev (Labs, 2024) | 12.0 | **0.67** | 0.99 | 0.81 | 0.79 | 0.74 | 0.20 | 0.47 |
| FLUX-schnell (Labs, 2024) | 12.0 | **0.71** | 0.99 | 0.92 | 0.73 | 0.78 | 0.28 | 0.54 |
| **Sana-0.6B** (Ours) | 0.6 | **0.64** | 0.99 | 0.76 | 0.64 | 0.88 | 0.18 | 0.39 |
| **Sana-1.6B** (Ours) | 1.6 | **0.66** | 0.99 | 0.77 | 0.62 | 0.88 | 0.21 | 0.47 |

Table 12: **Comparison of SOTA methods on DPG-Bench and ImageReward with details.** The table includes different metrics such as overall performance, entity, attribute, relation, and other categories.

| Model | Params (B) | Overall ↑ | Global | Entity | Attribute | Relation | Other | ImageReward ↑ |
| --- | --- | --- | --- | --- | --- | --- | --- | --- |
| **512 ×512 resolution** | | | | | | | | |
| PixArt-α (Chen et al., 2024b) | 0.6 | **71.6** | 81.7 | 80.1 | 80.4 | 81.7 | 76.5 | 0.92 |
| PixArt-Σ (Chen et al., 2024a) | 0.6 | **79.5** | 87.5 | 87.1 | 86.5 | 84.0 | 86.1 | 0.97 |
| **Sana-0.6B** (Ours) | 0.6 | **84.3** | 82.6 | 90.0 | 88.6 | 90.1 | 91.9 | 0.93 |
| **Sana-1.6B** (Ours) | 0.6 | **85.5** | 90.3 | 91.2 | 89.0 | 88.9 | 92.0 | 1.04 |
| **1024 ×1024 resolution** | | | | | | | | |
| LUMINA-Next (Zhuo et al., 2024) | 2.0 | **74.6** | 82.8 | 88.7 | 86.4 | 80.5 | 81.8 | - |
| SDXL (Podell et al., 2023) | 2.6 | **74.7** | 83.3 | 82.4 | 80.9 | 86.8 | 80.4 | 0.69 |
| PlayGroundv2.5 (Li et al., 2024a) | 2.6 | **75.5** | 83.1 | 82.6 | 81.2 | 84.1 | 83.5 | 1.09 |
| Hunyuan-DiT (Li et al., 2024c) | 1.5 | **78.9** | 84.6 | 80.6 | 88.0 | 74.4 | 86.4 | 0.92 |
| PixArt-Σ (Chen et al., 2024a) | 0.6 | **80.5** | 86.9 | 82.9 | 88.9 | 86.6 | 87.7 | 0.87 |
| DALLE3 (OpenAI, 2023) | - | **83.5** | 91.0 | 89.6 | 88.4 | 90.6 | 89.8 | - |
| SD3-medium (Esser et al., 2024) | 2.0 | **84.1** | 87.9 | 91.0 | 88.8 | 80.7 | 88.7 | 0.86 |
| FLUX-dev (Labs, 2024) | 12.0 | **84.0** | 82.1 | 89.5 | 88.7 | 91.1 | 89.4 | - |
| FLUX-schnell (Labs, 2024) | 12.0 | **84.8** | 91.2 | 91.3 | 89.7 | 86.5 | 87.0 | 0.91 |
| **Sana**-0.6B (Ours) | 0.6 | **83.6** | 83.0 | 89.5 | 89.3 | 90.1 | 90.2 | 0.97 |
| **Sana**-1.6B (Ours) | 1.6 | **84.8** | 86.0 | 91.5 | 88.9 | 91.9 | 90.7 | 0.99 |

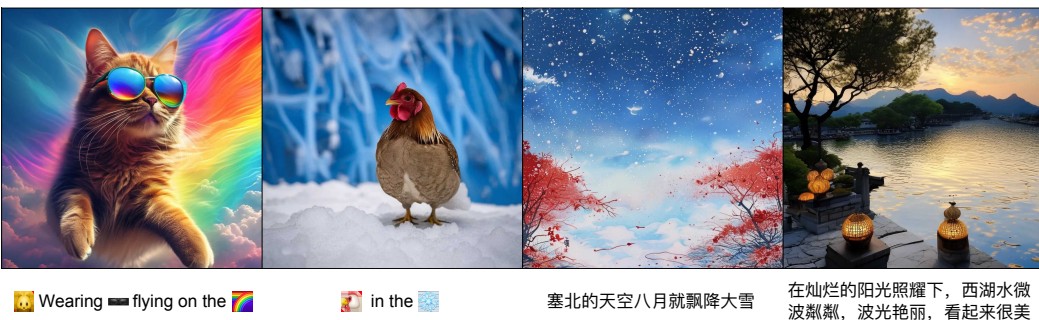

Figure 14: Visualization of zero-shot language transfer ability. Our Sana only has English prompts during training but can understand Chinese/Emoji during inference. This benefits from the generalization brought by the powerful pre-training of Gemma-2.

Table 13: **Performance of Sana block design space.** We train all the models with the same training setting with 52K iterations.

| Blocks | AE | Res. | FID ↓ | CLIP ↑ |
|---|---|---|---|---|
| FullAttn & FFN | F8C4P2 | 256 | 18.7 | 24.9 |
| + Linear | F8C4P2 | 256 | 21.5 | 23.3 |
| + MixFFN | F8C4P2 | 256 | 18.9 | 24.8 |
| + Kernel Fusion | F8C4P2 | 256 | 18.8 | 24.8 |
| Linear+GLUMBConv2.5 | F32C32P1 | 512 | 6.4 | 27.4 |
| + Kernel Fusion | F32C32P1 | 512 | 6.4 | 27.4 |

Table 14: Comparison of various T5 models and Gemma models based on speed and parameters. The sequence length (Seq Len) is the number of text tokens.

| Text Encoder | Batch Size | Seq Len | Latency (s) | Params (M) |
|---|---|---|---|---|
| T5-XXL | | | 1.6 | 4762 |
| T5-XL | | | 0.5 | 1224 |
| T5-large | | | 0.2 | 341 |
| T5-base | 32 | 300 | 0.1 | 110 |
| T5-small | | | 0.0 | 35 |
| Gemma-2b | | | 0.2 | 2506 |
| Gemma-2-2b | | | 0.3 | 2614 |

# D DISCUSSION OF POTENTIAL MISUSE OF SANA

Misusing generative AI models to generate NSFW content is a challenging issue for the community. To enhance safety, we have equipped SANA together with a safety check model (e.g. ShieldGemma-2B (Zeng et al., 2024)). Specifically, the user prompt will first be sent to the safety check model to determine whether it contains NSFW(not safe for work) content. If the user prompt does not contain NSFW, it will continue to be sent to Sana to generate an image. If the user prompt contains NSFW content, the request will be rejected. After extensive testing, we found that ShieldGemma can perfectly filter out NSFW prompts entered by users under strict thresholds and our pipeline will not create harmful AI-generated content.

Table 15: Comparison of throughput and latency under different resolutions. All models tested on an A100 GPU with FP16 precision.

| Methods | Speedup | Throughput(/s) | Latency(ms) | Methods | Speedup | Throughput(/s) | Latency(ms) |
|---|---|---|---|---|---|---|---|
| 512×512 Resolution | | | | 1024×1024 Resolution | | | |
| SD3 | 7.6x | 1.14 | 1.4 | SD3 | 7.0x | 0.28 | 4.4 |
| FLUX-schnell | 10.5x | 1.58 | 0.7 | FLUX-schnell | 12.5x | 0.50 | 2.1 |
| FLUX-dev | 1.0x | 0.15 | 7.9 | FLUX-dev | 1.0x | 0.04 | 23 |
| PixArt-Σ | 10.3x | 1.54 | 1.2 | PixArt-Σ | 10.0x | 0.40 | 2.7 |
| HunyuanDiT | 1.3x | 0.20 | 5.1 | HunyuanDiT | 1.2x | 0.05 | 18 |
| **Sana-0.6B** | 44.5x | 6.67 | 0.8 | **Sana-0.6B** | 43.0x | 1.72 | 0.9 |
| **Sana-1.6B** | 25.6x | 3.84 | 0.6 | **Sana-1.6B** | 25.2x | 1.01 | 1.2 |
| 2048×2048 Resolution | | | | 4096×4096 Resolution | | | |
| SD3 | 5.0x | 0.04 | 22 | SD3 | 4.0x | 0.004 | 230 |
| FLUX-schnell | 11.2x | 0.09 | 10.5 | FLUX-schnell | 13.0x | 0.013 | 76 |
| FLUX-dev | 1.0x | 0.008 | 117 | FLUX-dev | 1.0x | 0.001 | 1023 |
| PixArt-Σ | 7.5x | 0.06 | 18.1 | PixArt-Σ | 5.0x | 0.005 | 186 |
| HunyuanDiT | 1.2x | 0.01 | 96 | HunyuanDiT | 1.0x | 0.001 | 861 |
| **Sana-0.6B** | 53.8x | 0.43 | 2.5 | **Sana-0.6B** | 104.0x | 0.104 | 9.6 |
| **Sana-1.6B** | 31.2x | 0.25 | 4.1 | **Sana-1.6B** | 66.0x | 0.066 | 5.9 |

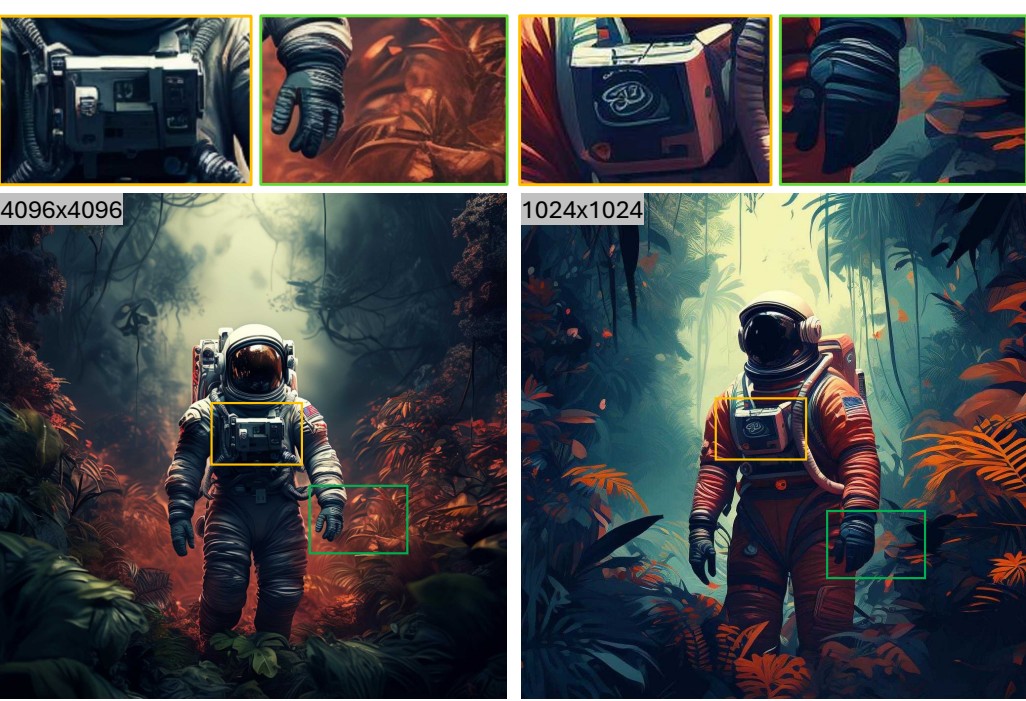

Figure 15: Comparison of 4K and 1K resolution images. We can see that the 4K image contains more details.

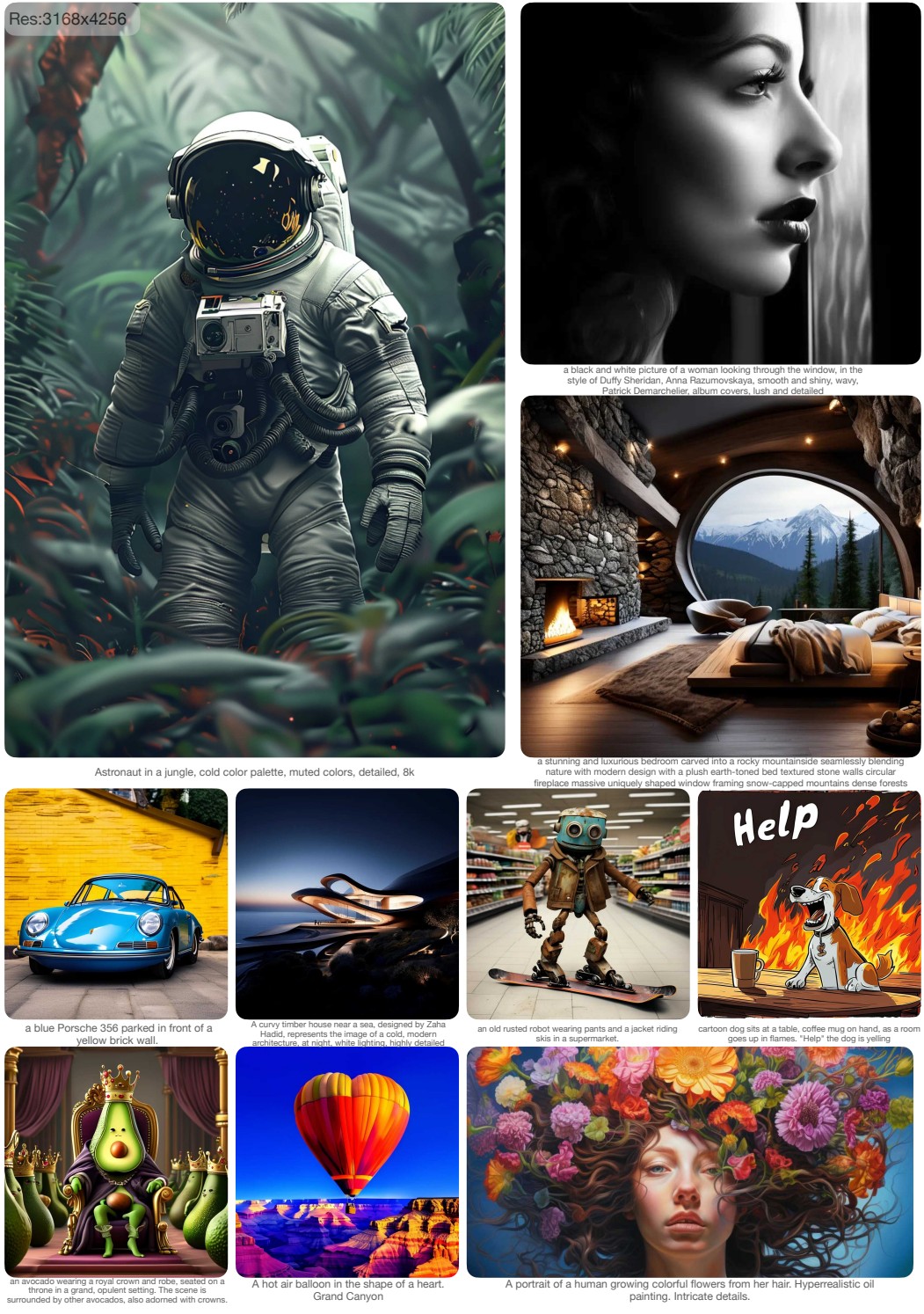

Figure 16: More samples generated from Sana.

