# OpenReview forum: "SANA: Efficient High-Resolution Text-to-Image Synthesis with Linear Diffusion Transformers"
_ICLR.cc/2025/Conference — ICLR 2025 Oral_

### Official Review · Reviewer_Dipw · 2024-10-29

**Soundness:** 3
**Presentation:** 3
**Contribution:** 2
**Rating:** 8
**Confidence:** 4

**Summary:**

The paper introduces Sana, a novel framework for efficient text-to-image generation, capable of producing 4K images while maintaining strong text-image alignment at remarkable speed. Sana’s core innovations include: a deep compression autoencoder, a Linear DiT architecture to enhance efficiency and a LLM-based text encoder. The model achieves competitive results compared to much larger models like Flux-12B, boasting significantly lower computational requirements.

**Strengths:**

1. Efficiency is a really big selling point. With a very high compression rate of deep autoencoder and linear DiT, SANA can generate high-quality images even on a laptop is very impressive.

2. The fact that using a small but good LLM can outperform a big but bad text encoder is another interesting point. This boosts text-image alignment a lot when used with their proposed complex human instruction (CHI) pipeline.

3. They provide extensive ablation experiments on each components and measure their impact on efficiency and perform of the model, which is very helpful for future work to build upon.

**Weaknesses:**

1. Although the results are impressive, the contribution of this paper on the technical side is somewhat limited. Their three main components have been already explored in the literature, deep autoencoder in [1], linear DiT in [2] and using LLM as text encoder in [3, 4]. So combining them is not really a significant novel idea.

2. In section 2.3, the author do not fully explain their CHI pipeline, is it same as [5] or completely different. If it is similar then this even reduces the novelty of the work further.

Reference:

[1] Chen et. al. Deep Compression Autoencoder for Efficient High-Resolution Diffusion Models. 2024

[2] Liu et. al. LinFusion: 1 GPU, 1 Minute, 16K Image. 2024

[3] Hu et.al. ELLA: Equip Diffusion Models with LLM for Enhanced Semantic Alignment. 2024

[4] Liu et.al. LLM4GEN: Leveraging Semantic Representation of LLMs for Text-to-Image Generation. 2024

[5] Ma et. al. Exploring the Role of Large Language Models in Prompt Encoding for Diffusion Models. 2024

**Questions:**

I do not have any further question

**Details Of Ethics Concerns:**

Because SANA is a powerful yet very efficient text-to-image model then the paper should include a paragraph to discuss about its potential misuse like creating harmful AI-generated contents and upload them online.

---

> ### Author Response · Authors · 2024-11-22
>
> **Thank you very much for taking the time to review and for your support. We try our best to address your questions as follows.**
>
> ### Q1: Difference with previously explored methods
>
> Thank you for your insightful comments regarding SANA combining several explored components, but SANA does not simply combine several existing components but makes significant modifications:
>
> **A1**: 1. Linear Attention in LinFusion[2] vs. SANA:
> - Differences in use: LinFusion employs linear attention mechanisms selectively and sparingly within its U-Net based structure (SDv2 and SDXL), mainly as a supplement to standard self-attention layers. This contrasts sharply with our approach in the DiT architecture, where linear attention is used comprehensively across all layers, enhancing computational efficiency in generative tasks.
> - Foundational vs. distillation Focus: LinFusion’s use of linear attention is part of a distillation strategy. In contrast, our model is designed as a foundational generative model and is trained from scratch, where the global replacement of vanilla self-attention with linear attention is explored in depth. We discovered through extensive experimentation that straightforward replacement does not suffice; it requires a suite of supportive design tweaks, such as the Mix-FFN as discussed in Section 2.2, to achieve optimal performance.
> - Technical implementation: The linear attention in LinFusion relies on matrix compression techniques akin to State Space Models (SSM), mapping all tokens to a reduced NxN low-rank state before updating. Our implementation, however, adheres more closely to direct O(N) attention computation, which is integral to maintaining the integrity of the transformer architecture. We aim to successfully apply the linear attention designs from the perception field to the generative field, which has not been effectively addressed in previous studies.
> 2. DC-AE[1]: First, DC-AE and SANA are concurrent works and both are submitted to ICLR 2025. Second, while higher compression ratio autoencoder is discussed in concurrent works DC-AE,  they focused more on ImageNet data reconstruction and simple class-condition generation. However, SANA focus on efficient and high-quality Text-to-Image generation, which presents unique challenges and requirements not addressed by DA-AE paper, as shown below:
>     - Directly replaced with DC-AE on our baseline PixArt-Sigma leading to unsatisfactory results. Our in-depth investigation found that letting DiT focus on denoising purely instead of compressing tokens can compensate for performance loss (e.g. patch size:  2 -> 1).
>     -  Simply enlarging the channel in latent space (F8C4->F32C32) will make training converge much slower, we instead design Linear Attention + Mix-FFN blocks for fast convergence
>     - We design complex human instruction and learnable small-scale factor~(e.g. initialize it with 0.01) in Decoder-only LLMs to stabilize training and improve semantic alignment.

---

> ### Author Response · Authors · 2024-11-22
>
> ### Q2. Difference usage of Decoder-only LLM
>
> **A2**: Both ELLA[3] and LLM4GEN[4] are early explorations of using LLM as a text encoder, we will cite them in our paper. However, they only simply replace T5/Clip with LLM and lack research on stimulating LLM's reasoning capabilities. LiDiT[5] attempts to promote the reasoning ability of LLM through simple human instruction, but it is a relatively simple attempt. Additionally, LiDiT is also necessary to fine-tune the last few layers of the LLM model, which increases the training cost and pipeline complexity. Our CHI in SANA designs more complex prompt templates and in-context-learning techniques, which can fully stimulate the high-order reasoning ability of LLM, extract better text embedding, and thus improve semantic alignment capabilities.
>
> We added the ablation study below to compare the effects of LiDiT and SANA's CHI and found that CHI consistently outperformed LiDiT.
>
> Complex Human Instruction(SANA) vs Simple Human Instruction(SHI)(LiDiT)
> | Train Step     | CHI        |  SHI         |
> | -------------- | ---------- | ------------ |
> | from 360K 	 |     -      |    -         |
> | +2K(finetune)  | 0.636      |  0.617       |
> | +5K(finetune)  | 0.642      |  0.626       |
>
> ### Q3 Discuss Potential Misuse of SANA
>
> Good question, this is a common question for all image generative AI models.
> To enhance safety, we have equipped SANA together with a safety check model (e.g. ShieldGemma-2B[6]). Specifically, the user prompt will first be sent to the safety check model to determine whether it contains NSFW(not safe for work) content. If the user prompt does not contain NSFW, it will continue to be sent to SANA to generate an image. If the user prompt contains NSFW content, the request will be rejected. After extensive testing, we found that ShieldGemma can perfectly filter out NSFW prompts entered by users under strict thresholds and our pipeline will not create harmful AI-generated content. We will add this part to our final paper.
>
>
> [1] Chen et. al. Deep Compression Autoencoder for Efficient High-Resolution Diffusion Models. 2024
>
> [2] Liu et. al. LinFusion: 1 GPU, 1 Minute, 16K Image. 2024
>
> [3] Hu et. al. ELLA: Equip Diffusion Models with LLM for Enhanced Semantic Alignment. 2024
>
> [4] Liu et. al. LLM4GEN: Leveraging Semantic Representation of LLMs for Text-to-Image Generation. 2024
>
> [5] Ma et. al. Exploring the Role of Large Language Models in Prompt Encoding for Diffusion Models. 2024
>
> [6] ShieldGemma: Generative AI Content Moderation Based on Gemma
>  https://huggingface.co/google/shieldgemma-2b

---

> ### Comment · Reviewer_Dipw · 2024-11-23
> **Regarding to the rebuttal**
>
> Thank the authors for the detail answers to my questions. Most of my concerns, especially about novelty, are well-addressed and I have to admit that I was wrong about DC-AE as I did not check that it was indeed a concurrent work. Also, is the Fig. 13 the description of how the authors perform CHI? The author should inlclude a brief paragraph saying that figure 13 is a description of CHI pipeline, so that it is easier for readers to search. Based on the rebuttal, I've increased **my score to 8**. Below I have two additional remarks (these are merely further suggestions and have no impact on my scoring):
>
> 1. regarding to the question about token limit of CLIP from Reviewer 9t32. The authors can consider to use AltCLIP [6] to evaluate CLIPScore as its token limit is 514 which I believe is long enough.
>
> 2. And the question raised by Reviewer S4E6 about evaluation for 4K is interesting. What evaluation protocol that the authors think is suitable for 4K ? Because there is an increasing number of work tackles the 4K image synthesis task.
>
> Reference:
> [6] https://github.com/FlagAI-Open/FlagAI/tree/master/examples/AltCLIP

---

> > ### Author Response · Authors · 2024-11-25
> >
> > Thank you for increasing the score and giving constructive suggestions!
> >
> > 1. Regarding the long-context clip model: Sure we will try this AltCLIP to score the text-image pairs and the VQScore mentioned by Reviewer 9t32. Thank you for providing it to us!
> >
> > 2. Regarding 4K image generation evaluation: There is actually a lack of research in this area. Some existing metrics (e.g. PSNR, SSIM) are suitable for evaluating 4K super-resolution, and some metrics are suitable for evaluating the alignment ability of text-to-image generation (e.g. GenEval, DPGBench). In addition, there is a lack of good evaluation of 4K datasets for fidelity quality assessment.
> > In general, I believe we should consider both 4K pixel quality evaluation and semantic alignment quality evaluation. Maybe collecting a 4K dataset like MJHQ is a good way to do so and we will definitely consider this test dataset in our following projects.

---

### Official Review · Reviewer_S4E6 · 2024-10-30

**Soundness:** 3
**Presentation:** 3
**Contribution:** 3
**Rating:** 8
**Confidence:** 4

**Summary:**

This paper proposes a text-to-image framework that can efficiently generate images up to 4096 x 4096 resolution. Comparing to existing works, the paper proposes four designs to facilitate fast speed, high quality text-to-image generation: (1) An auto encoder that down samples to input image by 32 times. (2) Replacing the quadratic self attention with linear attention. (3) Applies decoder-only LLM as text encoder. (4) Adopts efficient caption labelling and Flow-DPM-Solver for training and inference. Experimental results show that the proposed model achieves better or comparable results in terms of image quality, text image alignment comparing to existing methods, while having much higher throughput and lower latency

**Strengths:**

1. The paper is well motivated, as developing method that facilitates high quality, high resolution text-to-image generation in resource limited scenarios is of great application value.
2. The paper addresses the problem from several aspects, including auto encoder design, light weight self attention module, sampling method, and better caption labeling procedure.
3. The experimental results on resolutions of 512 x 512, 1024 x 1024 show that the proposed indeed achieve better or comparable performance comparing to previous state of the art methods, while having much higher throughput and lower latency.
4. The ablation experiments of different design components are sufficient and convincing.

**Weaknesses:**

1. The design of linear attention block needs to be explained in more detail. For example, the original self attention helps modelling long range dependency between tokens, here in the linear attention block, how is the dependency / contextual relation between tokens modeled through the new operations ?
2. The paper claims that the proposed method is able to generate images up to 4096 x 4096 resolution, however, in terms of quality comparison with existing methods, there are only ones with resolution of 512, 1024. For 4096 resolution, only the speed comparison is reported (Table 14), it would be more convincing to include FID and other image-text alignment metrics comparison for 4096 resolution as well.

**Questions:**

Please see weaknesses section

---

> ### Author Response · Authors · 2024-11-22
>
> **Thank you very much for taking the time to review and for your support. We try our best to address your questions as follows.**
>
> ### Q1: Details of linear attention modeling
>
> **A1**: Our linear attention maintains the global receptive field and long-range dependency capabilities of traditional vanilla self-attention. The primary modification involves replacing the softmax similarity computation with a ReLU-based similarity function[1,2]. This change allows us to leverage the associative property of matrix multiplication, significantly reducing computational complexity and memory usage from quadratic to linear, without compromising the mechanism’s effectiveness. Specifically:
>
> As discussed in EfficientViT[2], given input $x \in \mathbb{R}^{N \times f}$, traditional softmax attention can be expressed as:
>
> $$
> O_i = \sum_{j=1}^{N} \left[ \frac{Sim(Q_i, K_j)}{\sum_{j=1}^{N} Sim(Q_i, K_j)} \right] \cdot V_j
> \tag{1}
> $$
>
> where $Q = xW_Q$, $K = xW_K$, $V = xW_V$ are projections, and $Sim(Q, K) = \frac{\exp(QK^T)}{\sqrt{d}}$ defines the softmax-based similarity.
>
> To reduce computational complexity from quadratic to linear, we use ReLU linear attention:
>
> $$
> Sim(Q, K) = ReLU(Q)ReLU(K)^T,
> $$
>
> which modifies the output calculation to:
>
> $$
> O_i = \sum_{j=1}^{N} \frac{ReLU(Q_i)ReLU(K_j)^T}{\sum_{j=1}^{N} ReLU(Q_i)ReLU(K_j)^T} V_j
> $$
>
>
> $$
> = \frac{\sum_{j=1}^{N} (ReLU(Q_i)ReLU(K_j)^T)V_j}{ReLU(Q_i) \sum_{j=1}^{N} ReLU(K_j)^T}
> \tag{2}
> $$
>
> This formulation enables using the associative property of matrix multiplication, simplifying the calculation of $O_i$ to:
>
> $$
> O_i = \frac{\sum_{j=1}^{N} [ReLU(Q_i)ReLU(K_j)^T]V_j}{ReLU(Q_i) \sum_{j=1}^{N} ReLU(K_j)^T}
> $$
>
>
> $$
> = \frac{\sum_{j=1}^{N} ReLU(Q_i)[ReLU(K_j)^T V_j]}{ReLU(Q_i) \sum_{j=1}^{N} ReLU(K_j)^T}
> $$
>
>
> $$
> = \frac{ReLU(Q_i)(\sum_{j=1}^{N} ReLU(K_j)^T V_j)}{ReLU(Q_i)(\sum_{j=1}^{N} ReLU(K_j)^T)}
> \tag{3}
> $$
>
> As shown in Eq. (3), we only need to compute $(\sum_{j=1}^{N} ReLU(K_j)^T V_j) \in \mathbb{R}^{d \times d}$ and $(\sum_{j=1}^{N} ReLU(K_j)^T) \in \mathbb{R}^{d \times 1}$ once, then can reuse them for each query, thereby only requires $\mathcal{O}(N)$ computational cost and $\mathcal{O}(N)$ memory.
>
> Besides, We discovered through extensive experimentation that straightforward replacement of vanilla self-attention to linear attention does not suffice; it requires a suite of supportive design tweaks, such as the Mix-FFN as discussed in Section 2.2, to achieve optimal performance.
>
> [1] Angelos Katharopoulos, Apoorv Vyas, Nikolaos Pappas, and Fran ̧ cois Fleuret. Transformers are rnns: Fast autoregressive transformers with linear attention. In International Conference on Machine Learning, pages 5156–5165. PMLR, 2020.
>
> [2] Han Cai, Junyan Li, Muyan Hu, Chuang Gan, and Song Han. Efficientvit: Lightweight multi-scale attention for high-resolution dense prediction. In Proceedings of the IEEE/CVF International Conference on Computer Vision, pp. 17302–17313, 2023.
>
>
> ### Q2: Should include metrics e.g. FID on 4K image generation?
>
>
> **A2**: Thanks for the insightful asking. Currently, the datasets primarily utilized, such as COCO and MJHQ, consist of images up to 1K resolution, and there are no established versions of the 4K version of InceptionNet, which is typically used for FID calculations. Evaluating on 4K would not be particularly meaningful due to the lack of comparative methods and suitable benchmarks. Additionally, there is no standardized approach for conducting FID analysis at 4K resolution. Therefore, at this stage, our focus is on exploratory analysis with the available resolutions, primarily assessing visual quality rather than striving for high-resolution FID metrics.

---

> > ### Comment · Reviewer_S4E6 · 2024-11-24
> >
> > Thank you for the detailed response, I have increased my rating.

---

> > > ### Author Response · Authors · 2024-11-25
> > >
> > > Thank you so much for your support! We will keep moving to make Sana and the following projects better and better.

---

### Official Review · Reviewer_mBM3 · 2024-11-01

**Soundness:** 3
**Presentation:** 3
**Contribution:** 3
**Rating:** 8
**Confidence:** 4

**Summary:**

The paper "SANA: Efficient High-Resolution Image Synthesis with Linear Diffusion Transformers" introduces SANA, a high-efficiency text-to-image generation framework capable of producing high-resolution images up to 4096×4096 pixels. SANA employs multiple innovations, including a deep compression autoencoder (compressing images 32× for latency reduction), linearized Diffusion Transformer (DiT) attention to improve high-resolution efficiency, and the use of a decoder-only LLM as a text encoder with "Complex Human Instruction" for enhanced image-text alignment. The model also implements a Flow-DPM-Solver to reduce sampling steps, achieving competitive quality at 100× faster speeds compared to state-of-the-art diffusion models. SANA's improvements are demonstrated on high-resolution benchmarks and support deployment on a laptop GPU.

**Strengths:**

1. Innovative Efficiency Strategies: SANA’s integration of linear attention, a high-compression autoencoder, and a unique sampling solver enables fast high-resolution generation, which holds practical value for many applications.
2. Effective Use of Large Language Model (LLM) for Text Encoding: SANA’s implementation of a decoder-only LLM with CHI to refine prompts improves text-to-image alignment, enhancing quality without incurring high latency.

**Weaknesses:**

Limited Ablation in Model Design: While SANA combines existing methods effectively, many techniques—such as linear attention, high-compression auto-encoders, and Flow-based solvers—are iterative upon recent advancements in diffusion transformers. Explicit comparisons with recent models like PixArt-Σ or Playground v3, which also incorporate high-efficiency strategies or decoder-only LLMs, with component by component comparisons would better attribute SANA’s unique contributions.

**Questions:**

1. SANA uses the Gemma-2 LLM with Complex Human Instruction (CHI) to improve text-image alignment. While similar works like Playground v3 uses Llama3-8B, this paper only compared T5 with Gemma, do we know if larger LLMs can further improve the generation quality?

---

> ### Author Response · Authors · 2024-11-22
>
> **Thank you very much for taking the time to review and for your support. We try our best to address your questions as follows.**
>
> ### Q1: In detail comparison with other works e.g. PixArt-Sigma and Playgroundv3
>
>
> **A1**: We acknowledge the call for a detailed ablation study and provide extensive comparisons throughout the paper to highlight Sana’s advancements over existing technologies such as PixArt-Sigma. Actually, our experimental design and enhancements are systematically executed based on PixArt-Sigma, following a component-by-component approach to demonstrate Sana’s incremental and innovative contributions. Specifically,
> - VAE: Our high-compression autoencoder’s efficacy is demonstrated in Table 1 by comparing reconstruction abilities against PixArt-Sigma’s and SDXL’s VAE.
> - Training schedule: Additionally, in Table 3, we discuss the superiority of our training schedule based on flow-matching over PixArt-Sigma’s DDPM approach.
> - Captions: Table 4 highlights the benefits of our enhanced captioning approach with complementary VLMs and a temperature-based caption selection strategy using clip-score as outlined in Section 3.1, compared to PixArt-Sigma’s single caption strategy.
> - Model designs: Furthermore, our model design’s efficiency and capability improvements are evidenced in Tables 8 and 12 through innovations like Linear Attention, MixFFN, and Kernel-Fusion over the foundational model designs, Flash Attn & FFN, used by PixArt-Sigma.
>
> Regarding PlaygroundV3, direct comparisons are challenging due to differences in dataset compositions, unknown training scale, and the model’s size—24B for PlaygroundV3 vs. 1.6B/0.6B for Sana. These disparities make an apple-to-apple comparison difficult. Our primary focus is on efficiency, aiming to develop models that perform optimally on consumer-grade GPUs.
>
>
>
> ### Q2: Could enlarge LLMs further improve the generation quality?
>
> **A2**: Our results from Table 9 in the revised paper illustrate that larger models such as LLMs and DIT do indeed offer performance improvements. However, our focus remains on efficiency, particularly for deployment on consumer devices with 12/16GB of GPU memory. The next size up for Gemma is 9B, which would entail substantial GPU memory overhead. Training and inference of LLM models at bf16 precision require 18GB, which is impractical for consumer GPUs with less than 24GB for training and less than 16GB for inference. The costs associated with retraining a Gemma9B model are also prohibitively high. Therefore, we prioritize developing more efficiency-focused models that can deliver significant performance while remaining feasible for use on widely available hardware.

---

> > ### Comment · Reviewer_mBM3 · 2024-11-26
> >
> > Dear Authors,
> >
> > Thank you for the response, I have no further questions and will my rating.
> >
> > Best regards,

---

> > > ### Author Response · Authors · 2024-11-26
> > >
> > > Thank you so much for your kind words and strong support! We will keep moving to make Sana and the following projects better and better.

---

> ### Author Response · Authors · 2024-11-26
>
> Dear Reviewer mBM3,
>
> As the discussion period is nearing its conclusion, we kindly ask if you could review our response to ensure it addresses your concerns. Your feedback is greatly appreciated.
>
> Thank you for your time!
>
> Best,
>
> Authors

---

### Official Review · Reviewer_9t32 · 2024-11-01

**Soundness:** 4
**Presentation:** 4
**Contribution:** 3
**Rating:** 10
**Confidence:** 5

**Summary:**

The paper presents Sana, a cutting-edge text-to-image synthesis framework that can generate high-resolution images up to 4096×4096. Here are the key highlights:
Unlike conventional autoencoders that compress images by a factor of 8×, Sana's autoencoder achieves a remarkable 32× compression, drastically reducing the number of latent tokens.

Sana innovates by replacing the standard attention mechanisms in the Diffusion Transformer (DiT) with linear attention. This change enhances efficiency at high resolutions without sacrificing image quality.

Decoder-Only Text Encoder: Instead of relying on T5, Sana uses a modern, decoder-only small language model (LLM) as its text encoder. This model, combined with sophisticated human instructions and in-context learning, significantly improves text-image alignment.

Sana introduces Flow-DPM-Solver to cut down on the number of sampling steps and employs efficient caption labeling and selection techniques to speed up training convergence.

These advancements allow Sana to produce high-quality images at a significantly faster pace than existing models.

**Strengths:**

This work demonstrates its originality through several innovative contributions, including the Deep Compression Autoencoder, Linear DiT, and impressive 4K generation ability. These innovations enhance the quality and efficiency of high-resolution image generation while reducing computational requirements. The paper is well-written and clearly structured, with detailed experiments and results that validate the effectiveness of the proposed methods. Additionally, the significance of the work lies in its practical applications and the removal of limitations from prior models, making high-resolution image synthesis more accessible and scalable.
Overall, I find this work to be a breakthrough for diffusion text-to-image models.

**Weaknesses:**

1. The Gemma2-2B-IT model in Table 9 needs to be explained, as not all readers have a solid background in LLMs.

2. Further comparisons between Gemma2 and T5-XXL are needed, such as showing HPSv2 metrics on complex prompt benchmarks. The current FID scores in Table 9 are insufficient to highlight the true advantages of Gemma2.

3. Why does SANA perform significantly better than LUMINA-Next, even though both use the Gemma-2B model?

4. I believe some relevant works should be cited or compared, such as high-resolution generation work: UltraPixel: Advancing Ultra-High-Resolution Image Synthesis to New Peaks (NeurIPS 2024) and efficient generation work: Stable Cascade.

**Questions:**

1. The Gemma2-2B-IT model in Table 9 needs to be explained, as not all readers have a solid background in LLMs.

2. Further comparisons between Gemma2 and T5-XXL are needed, such as showing HPSv2 metrics on complex prompt benchmarks. The current FID scores in Table 9 are insufficient to highlight the true advantages of Gemma2.

3. Why does SANA perform significantly better than LUMINA-Next, even though both use the Gemma-2B model?

4. I believe some relevant works should be cited or compared, such as high-resolution generation work: UltraPixel: Advancing Ultra-High-Resolution Image Synthesis to New Peaks (NeurIPS 2024) and efficient generation work: Stable Cascade.

---

> ### Author Response · Authors · 2024-11-22
>
> **Thank you very much for taking the time to review and for your support. We try our best to address your questions as follows.**
>
>
> ### Q1 and Q4: Definition of Gemma2-2B-IT and adding more citations
> **A1 & A4**: Thanks a lot for your noticing. Gemma[1] is a family of lightweight, state-of-the-art open models from Google, built from the same research and technology used to create the Gemini models. The specific Gemma we use in Sana is the Gemma-2-2B-IT version. "IT" here represents the model is Instruct-Tuned for better prompt following and chat ability, experiments show that Instrucion-tuned versions are superior to non-Instruction-Tuned models, as shown in Table 9 in the revised version. The related papers mentioned by reviewers are also cited in the Appendix.
>
>
> ### Q2: More comprehensive comparison between Gemma and T5 series
>
> **A2**: Thank you for your insightful comments regarding the comparative evaluation of the Gemma2 and T5-XXL models as depicted in Table 9. As requested, we expanded our analysis beyond FID and CLIP-Score to include a comprehensive set of performance metrics from both GenEval and HPSv2 benchmarks, providing a broader perspective on each model’s capabilities.
>
> **Table 9: The extended results, as shown in the updated table below, highlight the comparative performance after 50K training steps on an A100 GPU with FP16 precision**
> | Text-Encoder   | #Params (M)|  Latency (s) | FID ↓   |  CLIP ↑ | GenEval ↑ | HPSv2 ↑ |
> | -------------- | ---------- | ------------ |-------- |-------- |---------- |-------- |
> | T5-XXL         | 4762       |  **1.61**       | 6.1     | 27.1    | **0.51**      | **27.95**   |
> | T5-Large       | 341        |  **0.17**        | 6.1     | 26.2    | **0.39**      | **27.57**   |
> | Gemma2-2B      | 2614       |  0.28        | 6.0     | 26.9    | 0.45      | 27.92   |
> | Gemma-2B-IT    | 2506       |  0.21        | 5.9     | 26.8    | 0.47      | 27.78   |
> | Gemma2-2B-IT   | 2614       |  **0.28**        | 6.1     | 26.9    | **0.50**      | **27.92**   |
>
> These results confirm that the Gemma2-2B-IT model not only matches the performance of the significantly larger and slower T5-XXL model in all metrics but does so with considerably higher efficiency. In addition, the Gemma-2B models outperform the T5-Large while operating at a similar speed, offering a more efficient solution.
>
> Furthermore, Gemma2 offers additional advantages including multilingual capabilities (zero-shot supporting Chinese and emoji), as detailed in Figure 13 of the Appendix, and features advanced in-context learning, illustrated in Figure 12. It is also optimized for rapid inference on consumer-grade GPUs with less than 12GB of memory, a capability demonstrated in the appendix video. These attributes make the Gemma2 series particularly suitable for applications requiring efficient, high-quality multilingual text processing on commonly available hardware.
>
>
> ### Q3: Why is Sana better than Lumina-Next with the same Gemma?
>
> **A3**: Thank you for your question regarding the comparative performance of Gemma2 versus Lumina-Next. It’s important to note that the language model is just one component of our approach, and there are several distinct aspects in which Gemma2 diverges from Lumina-Next:
>
> - 1. We use Gemma2-2B-Instruct, rather than Gemma1 used in Lumina-Next.
> - 2. Model Architecture: Sana features unique design elements such as a 32x compressed VAE and linear attention mechanisms, enhancing both its performance and efficiency, which is significantly different from Lumina-Next.
> - 3. Data Construction and Quality: We have enriched our dataset by integrating captions from multiple visual language models (VLMs) and employing a temperature-based caption selection strategy using clip-score. These methods not only diversify the dataset but also significantly enhance the model’s semantic alignment capabilities, as extensively discussed in Section 3.1.
>
>
> In summary, the above differences underpin the superior performance of Sana over Lumina-Next in key metrics, demonstrating its advanced semantic understanding capabilities.

---

> ### Comment · Reviewer_9t32 · 2024-11-23
> **Regarding the Experimental Results and the Use of ClipScore in the Paper**
>
> First and foremost, I would like to express my gratitude to the authors for their responses, which have addressed the concerns I previously raised. After re-examining the paper and the authors' latest replies, I have the following suggestions and questions:
>
> 1. I noticed that the authors have not updated the experimental results from their responses into the appendix of the paper. I believe these experimental findings are highly significant and valuable for reference, and they should be included in the paper, at the very least in the appendix.
>
> 2. The authors have emphasized the use of ClipScore. Upon reviewing the relevant section in the paper's appendix, I noticed that the authors did not explain the implementation or calculation method of ClipScore. I am curious about how ClipScore, which is based on CLIP and is known to have a text length limitation of approximately 77 tokens, is used to assess the quality of captions of such length.Regarding the Experimental Results and the Use of ClipScore in the Paper
>
> 3. I believe some relevant works should be cited, such as ultra-high-resolution generation: UltraPixel: Advancing Ultra-High-Resolution Image Synthesis to New Peaks (NeurIPS 2024) and efficient generation work: Stable Cascade.

---

> > ### Author Response · Authors · 2024-11-23
> >
> > **A5**: Thank you very much for your thoughtful and constructive feedback, as well as for your prompt response. We deeply appreciate the recognition of our efforts to address the concerns you’ve raised.
> >
> > We acknowledge the importance of including the updated experimental results in the paper and agree that these findings are significant. We commit to incorporating these results into the main and appendix of the revised version of the paper within the next 12 hours.
> >
> > In response to your query regarding our use of ClipScore, our approach primarily involves the selective strategy during the training process. We acknowledge that the 77-token limitation of CLIP may seem restrictive; however, we have found that this length is typically sufficient to encapsulate essential semantic information for our purposes. Consequently, we believe that the ClipScore based on 77 tokens still offers valuable insights and relevance for assessing caption quality in our context.

---

> > > ### Comment · Reviewer_9t32 · 2024-11-23
> > > **Regarding the use of ClipScore**
> > >
> > > Thank you for your detailed response regarding the use of ClipScore in your work. However, I remain concerned about the validity of your findings given the limitations of the 77-token constraint imposed by CLIP.
> > >
> > > To further investigate this issue, I conducted a simple experiment using the caption provided in your Figure 11 for the InternVL2-8B model (ClipScore: 26.37). The caption is as follows:
> > >
> > >
> > > >The image features a beautifully decorated chocolate cake with a heartfelt Valentine's Day message. The cake is round and sits on a dark, rustic wooden surface. The cake is covered in a rich, dark chocolate ganache, and it is adorned with a generous amount of chocolate shavings and crushed nuts, giving it a luxurious and indulgent appearance. At the top of the cake, the words "HAPPY VALENTINE" are written in bold, red icing, creating a striking contrast against the dark chocolate. The icing is piped in a slightly uneven, handwritten style, adding a personal touch to the message. Surrounding the cake are several red heart-shaped decorations, some of which are whole and others that are broken into smaller pieces, scattered around the cake. These hearts add to the romantic and festive theme of the cake. The cake is placed on a dark, round cake board, which complements the rustic wooden background. The overall presentation is elegant and festive, making it perfect for a Valentine's Day celebration.
> > >
> > > Using the CLIP tokenizer, I calculated the token length of this caption:
> > >
> > > ```
> > > from transformers import CLIPTokenizer
> > > tokenizer = CLIPTokenizer.from_pretrained("openai/clip-vit-base-patch32")
> > > tokens = tokenizer(Caption_InternVL2_8_caption, truncation=False)
> > > token_length = len(tokens.input_ids)
> > >
> > > print(f"Token length: {token_length}")
> > > ```
> > >
> > > The result was:
> > > ```
> > > Token length: 200
> > > ```
> > >
> > >
> > > This clearly indicates that the caption is significantly longer than the 77-token limit imposed by CLIP. Given this discrepancy, it is evident that the ClipScore you report is likely based only on the initial portion of the caption, which may not fully capture the semantic richness and detail of the entire description.
> > >
> > > In light of this, I have to question the reliability of your findings. If the ClipScore is indeed calculated using only the first 77 tokens, it may not accurately reflect the quality of the entire caption, potentially leading to misleading conclusions.
> > >
> > > I look forward to your response and hope to see a more robust evaluation methodology in your revised submission.

---

> > > > ### Comment · Reviewer_9t32 · 2024-11-23
> > > > **Regarding the use of ClipScore**
> > > >
> > > > Dear Authors,
> > > >
> > > > Please note that while I find the use of ClipScore to measure caption quality somewhat perplexing, **I do not consider this a major flaw or issue. I am highly appreciative of your primary technical contributions, and I simply hope that you can provide a clear explanation regarding the use of ClipScore in your work.**
> > > >
> > > > Best regards,

---

> ### Author Response · Authors · 2024-11-23
>
> **Thank you very much for your insightful questions and valuable suggestions.**
>
> Here, we try to address your concern on the caption selecting strategy based on CLIP-Score.
>
> **A6**: We primarily uses the CLIP model to score the caption as the selective strategy during training.
> The 77-token limitation of the original CLIP model may seem restrictive.
> However, after extensive observation, we found that the most valuable content typically appears in the first half of the caption, such as "The image captures **\$content of image\\$**...", that is, 77-token length is typically sufficient to encapsulate essential semantic information for our purposes. Consequently, we believe that the CLIP-Score based on 77 tokens still offers valuable insights and relevance for assessing caption quality in our context, which will benefit for performance rather than harmful.
>
> Additionally, we conducted an analysis of the effects of employing different sampling strategies. As illustrated in the figure (Caption-selecting-statistic.jpg) in the updated supplementary material zip file, using a ‘random’ temperature involves selecting a training sample randomly from various VLM captions. A temperature of 0.1 corresponds to sampling based on Clip-Score with this specific temperature setting, which moderates selection intensity, whereas a temperature of 0 always selects the caption with the highest Clip-Score.
>
> Upon analyzing 50,000 samples and the frequency with which three different captions were sampled, we observed that the strategies with temperatures of 0.1 and 0 consistently outperform the random selection approach. However, exclusively using the highest Clip-Score caption can lead to the issues noted by reviewers, where Clip-Score may not perfectly assess the image-text alignment. Consequently, we opted for a more balanced approach by setting the temperature to 0.1, aiming to select captions that more accurately reflect the semantic content of the images from a holistic dataset perspective.
>
> In the future, we will attempt to fine-tune a larger context-length CLIP model or use more advanced VLM to score the text-image pairs to make the data pipeline better.

---

> ### Comment · Reviewer_9t32 · 2024-11-24
> **Good Response and Further Suggestions**
>
> Dear Authors,
>
> Thank you for your detailed response to the concerns raised regarding the CLIP-Score caption selection strategy. I appreciate your clarification on the observation that "the most valuable content typically appears in the first half of the caption." While this point has been noted in previous work, your explanation does help to address the doubts about the use of CLIP-Score.
>
> However, I believe that VQASCore (ECCV'24), which directly measures text-image alignment using the T5-xxl model, could be a more reliable metric for assessing caption quality. VQASCore offers a more accurate assessment of text-image alignment.
>
> Therefore, I recommend considering the use of VQASCore or other advanced VLMs in future work to further optimize your data pipeline and enhance the performance and robustness of your model.
>
> Thank you once again for your efforts and detailed response.
>
> Best regards,

---

> > ### Comment · Reviewer_9t32 · 2024-11-24
> > **By the way**
> >
> > Dear Authors,
> >
> > By the way, I no longer see any obvious shortcomings in this paper. Therefore, I have increased my rating. This work should indeed be considered a highlight at the conference! Excellent job!
> >
> > Best regards,

---

> ### Author Response · Authors · 2024-11-25
>
> Thank you so much for your kind words and strong support! We will keep moving to make Sana and the following projects better and better. And of course, the VQScore and the AltCLIP[1] mentioned by the reviewer Dipw24 will be definitely considered in our dataset filter pipeline in the future.
>
> [1] https://github.com/FlagAI-Open/FlagAI/tree/master/examples/AltCLIP

---

### Author Response · Authors · 2024-11-22
**Response to all reviewers**

We sincerely thank all reviewers for their thorough reviews and constructive feedback. We are encouraged that the reviewers recognize SANA's innovations in efficient high-resolution image generation, practical value for resource-limited scenarios, and significant performance improvements while maintaining lower computational requirements (all reviewers).

## Key Improvements and Clarifications

1. Technical Novelty and Component Analysis
- Provided detailed comparisons with recent works like PixArt-Σ
- Demonstrated unique innovations in VAE design, training schedule, and model architecture
- Explained how SANA significantly improves upon existing techniques through Linear Attention Mix-FFN, LLM and CHI designs

2. Extended Evaluation and Analysis
- Added comprehensive metrics comparing Gemma2 and T5-XXL using GenEval and HPSv2
- Explained linear attention's efficiency while maintaining global receptive field
- Clarified evaluation challenges at 4K resolution due to benchmark limitations

3. Safety and Practical Considerations
- Integrated ShieldGemma for content moderation to prevent misuse
- Focused on consumer GPU compatibility and efficiency

Please see our detailed responses below each review and all the changes in the revised paper is high-lighted with orange font. We appreciate your feedback and welcome any additional questions.

---

> ### Author Response · Authors · 2024-11-23
>
> In the revised paper,
>
> 1. We add GenEval and HPSv2 to further compare T5 vs Gemma in Line 281 and Table 9. Asked by Reviewer-9t32.
> 2. We add more related works  in the paper, like, Stable Cascade, UltraPixel in Line 823.  Asked by Reviewer-9t32
> 3. We discuss the CLIP text encoder's input token length limitation asked by Reviewer-9t32 in Line 1021. Asked by Reviewer-9t32
> 4. We add more details about Gemma2-2B-IT used in Sana in Line 1041. Asked by Reviewer-9t32
> 5. We add more comparison with LiDiT's human instruction in Line 1059,1075, and add ablation in Table 10. Asked by Reviewer-Dipw
> 6. We discuss the potential misuse of AI models in Line 1221. Asked by Reviewer-Dipw
>
> Please see our detailed responses below each review for other questions and suggestions. We sincerely thank all reviewers for their thorough reviews and constructive feedback again.

---

### Meta-Review · Area_Chair_Z7Zi · 2024-12-16

**Metareview:**

This work presents SANA, a cutting-edge text-to-image synthesis framework that can generate high-resolution images up to 4096×4096.

It has the integration of linear attention, a high-compression autoencoder, and a unique sampling solver that enables fast, high-resolution generation, which holds practical value for many applications. Moreover, it also shows the effective use of the Large Language Model (LLM) for text encoding, which is simple yet effective. Extensive experiments on various Gen benchmarks show the effectiveness of SANA.

All reviewers feel positive before the rebuttal, with most concerns about more extensive experiment results and ablation studies.
After the rebuttal and discussion stage, all the issues are solved, and all reviewers raise their scores.

The area chair checks the refined version of this submission and suggests the authors merge the discussion results in their camera-ready version.

In addition, the author should also include a paragraph to discuss the potential misuse of SANA like creating harmful AI-generated contents in their final draft.

In the end, due to the effectiveness of the model and the practical potential of efficient generation, the area chair recommends this work as an oral presentation.

**Additional Comments On Reviewer Discussion:**

All the issues raised by reviewers are well solved.

---

### Decision · Program_Chairs · 2025-01-22

Accept (Oral)